# BLOS2 negatively regulates Notch signaling during neural and hematopoietic stem and progenitor cell development

Wenwen Zhou[1,2†], Qiuping He[2,3†], Chunxia Zhang[2,3], Xin He[1], Zongbin Cui[4], Feng Liu[2,3*], Wei Li[1,5,6,7,8*]

[1]State Key Laboratory of Molecular Developmental Biology, Institute of Genetics and Developmental Biology, Chinese Academy of Sciences, Beijing, China; [2]University of Chinese Academy of Sciences, Beijing, China; [3]State Key Laboratory of Membrane Biology, Institute of Zoology, Chinese Academy of Sciences, Beijing, China; [4]State Key Laboratory of Freshwater Ecology and Biotechnology, Institute of Hydrobiology, Chinese Academy of Sciences, Beijing, China; [5]Center for Medical Genetics, Beijing Children's Hospital, Capital Medical University, Beijing, China; [6]Center of Alzheimer's Disease, Beijing Institute for Brain Disorders, Beijing, China; [7]MOE Key Laboratory of Major Diseases in Children, Beijing, China; [8]Beijing Pediatric Research Institute, Beijing, China

*For correspondence: liuf@ioz.ac. cn (FL); liwei@bch.com.cn (WL)

[†]These authors contributed equally to this work

Competing interests: The authors declare that no competing interests exist.

**Abstract** Notch signaling plays a crucial role in controling the proliferation and differentiation of stem and progenitor cells during embryogenesis or organogenesis, but its regulation is incompletely understood. BLOS2, encoded by the *Bloc1s2* gene, is a shared subunit of two lysosomal trafficking complexes, biogenesis of lysosome-related organelles complex-1 (BLOC-1) and BLOC-1-related complex (BORC). *Bloc1s2*$^{-/-}$ mice were embryonic lethal and exhibited defects in cortical development and hematopoiesis. Loss of BLOS2 resulted in elevated Notch signaling, which consequently increased the proliferation of neural progenitor cells and inhibited neuronal differentiation in cortices. Likewise, ablation of *bloc1s2* in zebrafish or mice led to increased hematopoietic stem and progenitor cell production in the aorta-gonad-mesonephros region. BLOS2 physically interacted with Notch1 in endo-lysosomal trafficking of Notch1. Our findings suggest that BLOS2 is a novel negative player in regulating Notch signaling through lysosomal trafficking to control multiple stem and progenitor cell homeostasis in vertebrates.

## Introduction

Notch signaling is a highly conserved cell-to-cell signaling pathway and its function in cell-fate determination makes it essential for embryogenesis and organogenesis, including neurogenesis (*Traiffort and Ferent, 2015*), vasculogenesis (*Krebs et al., 2000*) and somite segregation (*Wright et al., 2011*). Cortical neurogenesis requires the exquisite coordination of neural progenitor cell (NPC) proliferation and differentiation to generate the complex and functional cerebral cortex, which is precisely regulated in temporal and spatial patterns (*Gal et al., 2006*; *Guillemot, 2005*). Notch signaling plays a crucial role in the control of proliferation of NPCs and neuronal differentiation during corticogenesis (*Fortini, 2009*; *Kopan and Ilagan, 2009*; *Pierfelice et al., 2011*). Notch signals activate the expression of basic helix loop helix (bHLH) Hes transcription factors that maintain undifferentiated NPCs by downregulating proneural gene expression (*Bertrand et al., 2002*; *Kageyama et al., 2007*; *Ohtsuka et al., 2001*; *Ross et al., 2003*). The neurogenic transition occurs

through stable downregulation of Notch target genes, leading to upregulation of proneural genes and neuronal differentiation (*Kawaguchi et al., 2008*; *Shimojo et al., 2008*). Hyperactivation of Notch signaling, such as that due to loss of Numb and Numblike, results in neural progenitor hyper-proliferation and impaired neuronal differentiation in the mouse brain (*Hoeck et al., 2010*; *Li et al., 2003a*; *Rodriguez et al., 2012*).

Moreover, the Notch signaling pathway has emerged as a key regulator in definitive hematopoiesis. It regulates artery/vein specification (*Lawson et al., 2002*), as well as hemogenic endothelium (HE) or hematopoietic stem and progenitor cell (HSPC) specification through downstream factors cell-autonomously (*Guiu et al., 2014*; *Jang et al., 2015*) or non-cell-autonomously (*Clements et al., 2011*; *Kobayashi et al., 2014*). Therefore, tight spatio-temporal regulation of Notch activity that determines its location, strength and duration is essential for the homeostasis and differentiation of NPCs or HSPCs. Current studies have focused primarily on the activation of Notch in animals. However, the precise control of Notch turnoff is largely unknown but extremely important in the orchestration of the Notch signals in development.

Notch receptor trafficking has been regarded as an important element in the regulation of Notch signaling (*Brou, 2009*; *Fortini and Bilder, 2009*). The endocytic trafficking of Notch receptor results in either transportation to lysosomes for degradation via multi-vesicular bodies (MVBs) and late endosomes or recycling back to the plasma membrane for ligand binding and activation (*Le Borgne, 2006*). When endocytic trafficking of Notch receptor destined for lysosomal degradation is disrupted, Notch receptor accumulates in endosomes and undergoes proteolytic cleavage and ectopic activation in a ligand-independent manner (*Furthauer and Gonzalez-Gaitan, 2009*; *Moberg et al., 2005*; *Thompson et al., 2005*; *Vaccari and Bilder, 2005*; *Vaccari et al., 2008*). In *Drosophila*, Deltex interacts with another E3 ubiquitin ligase, Su(Dx), to activate ligand-independent Notch proteolysis and signaling (*Cornell et al., 1999*). The HOPS and AP-3 complex are required for the Deltex-regulated activation of Notch in the endosomal trafficking pathway (*Wilkin et al., 2008*). Beyond flies, several mammalian proteins have been identified as regulators of Notch lysosomal degradation through the vacuolar H(+) ATPase (*Faronato et al., 2015*; *Kobia et al., 2014*; *Lange et al., 2011*; *Sethi et al., 2010*). However, additional regulators that are involved in Notch endocytic trafficking remain to be elucidated.

BLOS2 (encoded by the *Bloc1s2* gene) is a subunit of biogenesis of lysosome-related organelles complex-1 (BLOC-1), which has been reported to function in endo-lysosomal trafficking and in the biogenesis of lysosome-related organelles (LRO) (*John Peter et al., 2013*; *Setty et al., 2007*; *Starcevic and Dell'Angelica, 2004*; *Wei and Li, 2013*). A recent report reveals that BLOS2 is also a subunit of BLOC-1-related complex (BORC), which regulates the positioning of lysosomes (*Pu et al., 2015*). In addition, BLOS2 is likely to be associated with the centrosome to function in regulating transcription (*Sun et al., 2008*). Thus, BLOS2 might be a multi-functional protein and involved in regulating several cellular processes.

Several subunits of BLOC-1, such as dysbindin, snapin and BLOS1, mediate the transport of membrane receptors, including dopamine receptor 2 (D2R), NMDA receptor subtype 2A (NR2A), and epidermal growth factor receptor (EGFR), from endosomes to lysosomes for degradation (*Cai et al., 2010*; *Ji et al., 2009*; *Marley and von Zastrow, 2010*; *Tang et al., 2009*; *Zhang et al., 2014a*). Mice that lack BLOS1 or snapin are embryonic lethal, suggesting that these two subunits of BLOC-1 might play pivotal roles in embryonic development (*Tian et al., 2005*; *Zhang et al., 2014a*). BLOS2, together with BLOS1 and snapin, is a shared subunit of BLOC-1 and BORC (*Langemeyer and Ungermann, 2015*; *Pu et al., 2015*). Whether *Bloc1s2* knockout mice are embryonic lethal has not been reported. In addition, how BLOS2 functions in endo-lysosomal trafficking has not been clearly defined. Herein, we provide evidence to show that BLOS2 is a novel negative regulator of Notch signaling mediated by lysosomal trafficking, which is critical for NPC or HSPC development in vertebrates including zebrafish and mouse.

## Results

### BLOS2 is required for embryonic cortical morphogenesis and neurogenesis

To study the function of BLOS2 , we generated *Bloc1s2* knockout mice by replacing exons 1–4 of the *Bloc1s2* gene with the phosphoglycerate kinase-Neo (PGK-Neo) cassette (*Figure 1—figure supplement 1A*). By genotyping and immunoblotting, we confirmed the replacement of exons 1–4 and null BLOS2 protein in multiple tissues of *Bloc1s2*$^{-/-}$ neonate mice, while BLOS2 was highly expressed in brain, spleen and intestine in wild-type (WT) neonate mice (*Figure 1—figure supplement 1B and C*).

Genotyping of 1-week-old progeny derived from crossing *Bloc1s2*$^{+/-}$ mice revealed the absence of *Bloc1s2*$^{-/-}$ animals, implying that *Bloc1s2*$^{-/-}$ mice were mostly embryonic lethal. Very few homozygous mutant pups were born alive, and these died within a few hours of birth. This feature is similar to *Bloc1s1*$^{-/-}$ and snapin knockout mice (*Tian et al., 2005*; *Zhang et al., 2014a*) (*Supplementary file 1*), suggesting that the three shared subunits of BLOC-1 and BORC are essential for embryonic development. *Bloc1s2*$^{-/-}$ embryos or neonates manifested a variety of developmental defects, including impaired hematopoiesis (E12.5) (*Figure 1A*), loss of eye pigmentation (E14.5) (*Figure 1B*), craniofacial malformation (P0) (*Figure 1C*), and smaller brain size (E14.5) (*Figure 1D*).

We next focused on the phenotypes in brain development and hematopoiesis. Histological analyses revealed severe morphological defects in the developing cerebral cortex of *Bloc1s2*$^{-/-}$ embryos. Nissl staining of brains prepared from E14.5 *Bloc1s2*$^{-/-}$ embryos showed a thinner cerebral cortex as well as enlarged lateral ventricles (*Figure 1E and F*). Further examination of the cortical structure under higher magnification revealed that the *Bloc1s2*$^{-/-}$ cortical plate (CP) and intermediate zone (IZ) were remarkably thinner than WT, while the thickness of the ventricular zone (VZ) was comparable between WT and mutant (*Figure 1G and H*). Due to the significant reduction in cortical wall thickness, we next investigated the effect of loss of BLOS2 on early cortical neurogenesis by immunostaining with the deep-layer-specific markers Ctip2 (layer V and VI) and Tbr1 (layer VI) (*Lakomá et al., 2011*). At E12.5 and E14.5, Ctip2$^+$ early-born layer V and VI neurons and Tbr1$^+$ layer VI neurons were all significantly reduced in number in the *Bloc1s2*$^{-/-}$ cortex compared to the control (*Figure 1I* to *Figure 1K*), suggesting that BLOS2 ablation impairs early cortical neurogenesis. Taken together, these observations suggest that loss of BLOS2 disrupts proper cortical neurogenesis and morphogenesis.

### BLOS2 regulates the proliferation and differentiation of NPCs

To further investigate the pathogenesis of the thinner cortex of *Bloc1s2*$^{-/-}$ mice and the severe impairment of cortical neurogenesis in the *Bloc1s2*$^{-/-}$ cortex, we analyzed apoptosis, differentiation, and proliferation during early cortical development. We performed the terminal deoxynucleotidyl-transferase-mediated dUTP end-labeling (TUNEL) assay to assess cell death in WT and *Bloc1s2*$^{-/-}$ mice (*Li et al., 2003*). No significant change was observed in TUNEL signals in *Bloc1s2*$^{-/-}$ mice at E14.5, suggesting that the decrease in cortical neurons of *Bloc1s2*$^{-/-}$ mice does not arise from apoptotic cell death (*Figure 2A and B*). To determine whether *Bloc1s2*$^{-/-}$ mice display neuronal differentiation defects, we examined neurogenesis using the bromodeoxyuridine (BrdU)/Ki67 cell-cycle exit assay (*Siegenthaler et al., 2009*). BrdU was injected into pregnant dams at E13.5, and embryos were collected 18 hr later and processed for BrdU/Ki67 immunostaining. Cells remaining in cycle are BrdU$^+$ Ki67$^+$, whereas cells exited from cell cycling are BrdU$^+$ only. At E14.5, fewer BrdU$^+$ Ki67$^-$ cells left the cell cycle in *Bloc1s2*$^{-/-}$ mice compared to WT mice (*Figure 2C and D*). In addition, there was a significant reduction of Tuj1$^+$ early differentiated neurons in *Bloc1s2*$^{-/-}$ compared to WT mice (*Li et al., 2003*) (*Figure 2E and F*), suggesting impaired early neuronal differentiation of NPCs in the *Bloc1s2*$^{-/-}$ cortex.

We next explored whether proliferation of NPCs is altered in the *Bloc1s2*$^{-/-}$ cortex. We counted the number of Pax6$^+$ radial glia/apical progenitors in *Bloc1s2*$^{-/-}$ and WT animals. Pax6 immunostaining on coronal sections from E14.5 brains revealed a 52% increase in Pax6$^+$ cells in the mutant when compared with the WT, suggesting that the functional loss of BLOS2 results in an increase in the proliferation of Pax6$^+$ radial glia cells (*Figure 2G and H*). Moreover, we administered a 2-hr BrdU pulse

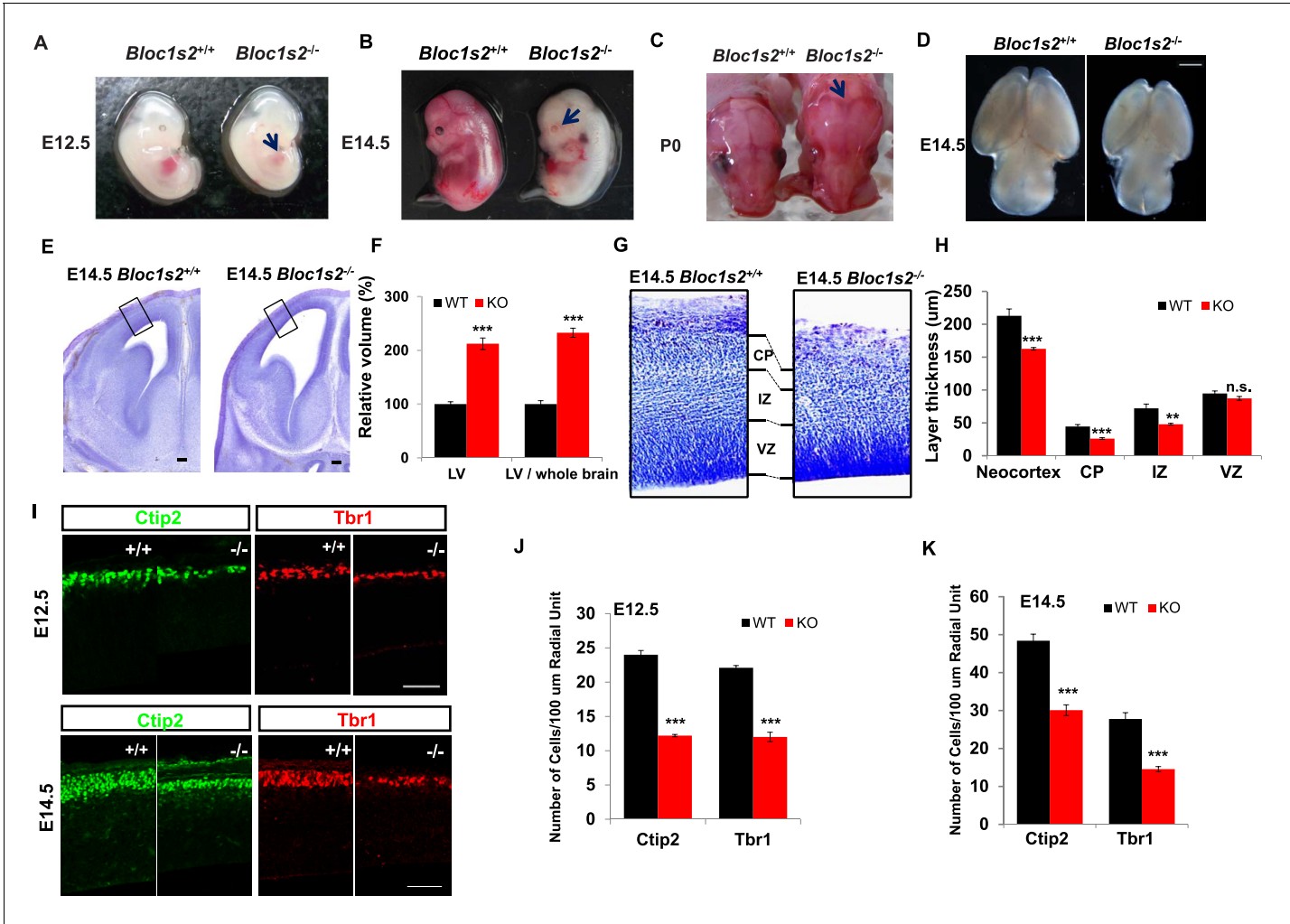

**Figure 1.** BLOS2-deficiency impaired cortical development. (A–D) Gross examination of whole mount WT (*Bloc1s2*^+/+) and *Bloc1s2*^−/− embryos at E12.5, E14.5, and newborns (P0). Note that the (A) impaired hematopoiesis, (B) loss of eye pigmentation, (C) craniofacial malformation, and (D) smaller brain size of *Bloc1s2*^−/− mice are indicated by the arrows compared with WT littermates at E12.5 and E14.5 and in newborns. (E) Coronal sections of E14.5 brains with Nissl staining of WT and *Bloc1s2*^−/− embryos. The black boxes indicate the areas shown at higher magnification in (G). The enlarged lateral ventricles (LV) are shown. (F) Volumetric analysis of the LV in WT and *Bloc1s2*^−/− brains at E14.5. The values for LV volume in *Bloc1s2*^−/− mice are indicated as percentages of the corresponding WT values. The ratios of the volume of LV to brain are also indicated as percentages of the corresponding WT values. ***p<0.001 (Student's t-test); n=6 mice. (G) The cortical plate (CP), intermediate zone (IZ), and ventricular zone (VZ) are labeled and compared. In *Bloc1s2*^−/− brain, the thinner CP and IZ are shown, but VZ is almost normal. (H) Quantification of the thickness of the neocortex, CP, IZ and VZ in WT and *Bloc1s2*^−/− brains at E14.5. 212.85 ± 10.39 (neocortex), 44.48 ± 3.01 (CP), 72.12 ± 6.46 (IZ), 94.66 ± 3.78 (VZ) for WT; 162.61 ± 1.99 (neocortex), 26.12 ± 1.27 (CP), 48.00 ± 1.26 (IZ), 87.17 ± 2.95 (VZ) for *Bloc1s2*^−/−. **p<0.01 (IZ); ***p<0.001 (neocortex and CP); n.s., not significant, p=0.137 (VZ); n=6 mice. (I) Immunohistochemistry (Ctip2 and Tbr1) on sections of WT and *Bloc1s2*^−/− brains at E12.5 and E14.5. The numbers of Ctip2^+ and Tbr1^+ cells were decreased in the *Bloc1s2*^−/− brain at E12.5 and E14.5. (J and K) Quantification of neuronal number at E12.5 and E14.5. 24.00 ± 0.62 (Ctip2^+), 22.08 ± 0.33 (Tbr1^+) for WT at E12.5, 12.22 ± 0.17 (Ctip2^+), 12.00 ± 0.68 (Tbr1^+) for *Bloc1s2*^−/− at E12.5; 48.40 ± 1.73 (Ctip2^+), 27.78 ± 1.66 (Tbr1^+) for WT at E14.5, 30.1 ± 1.41 (Ctip2^+), 14.55 ± 0.66 (Tbr1^+) for *Bloc1s2*^−/− at E14.5. ***p<0.001; n=9. All graphs are mean ± s.e.m. Scale bars: 1 mm in (D); 100 μm in (E); 50 μm in (I).

The following source data and figure supplements are available for figure 1:

**Figure supplement 1.** Generation of *Bloc1s2* knockout mice.

**Figure supplement 1—source data 1.** Western blots of *Figure 1—figure supplement 1C*.

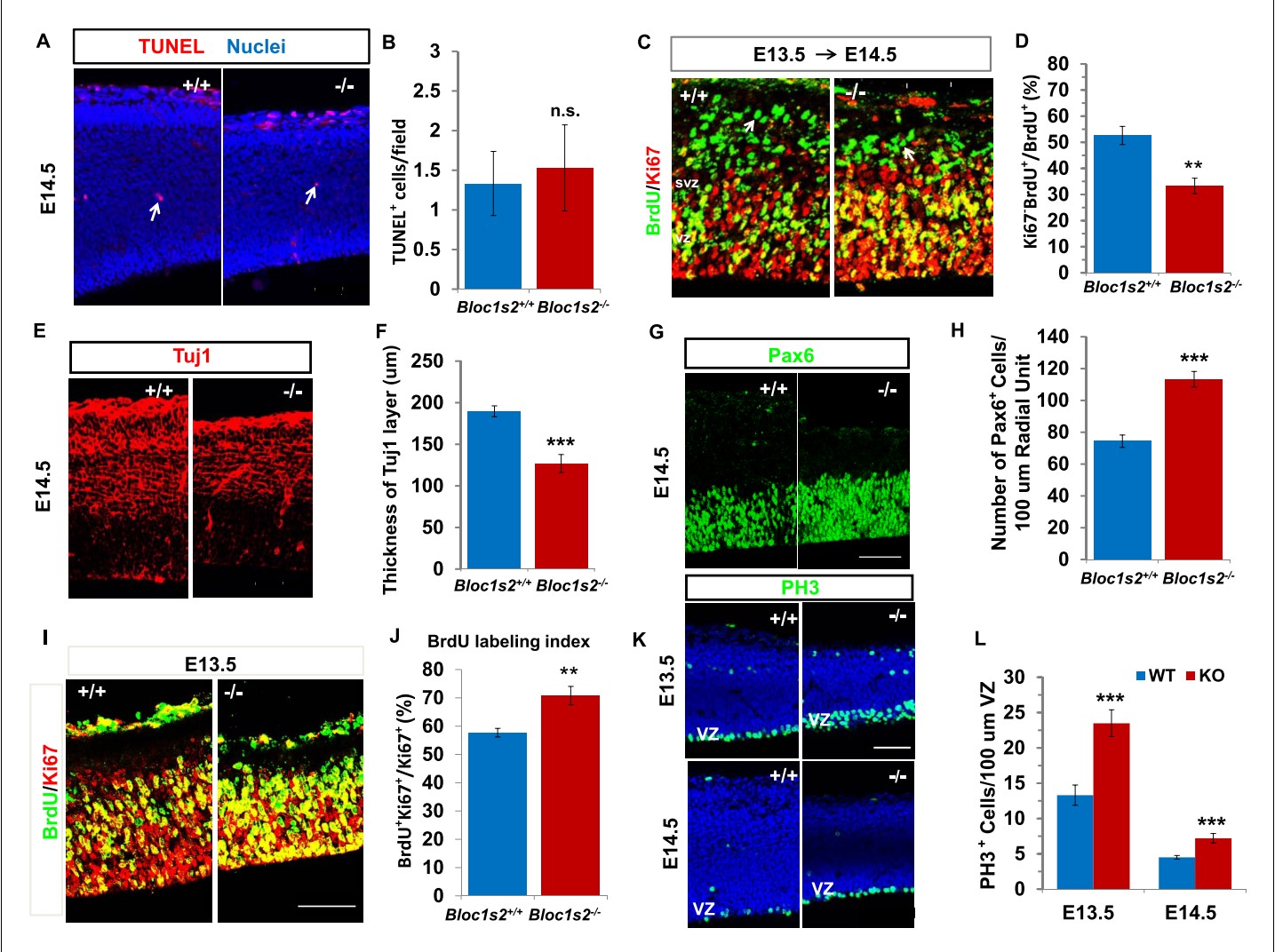

**Figure 2.** Neural progenitor proliferation and neurogenesis were altered in the $Bloc1s2^{-/-}$ cortex. (**A**) TUNEL assay on sections from WT and $Bloc1s2^{-/-}$ embryos at E14.5. No significant difference in the number of apoptotic cells (arrows) in the $Bloc1s2^{-/-}$ neocortex was detected as compared with WT littermates. (**B**) Quantification of TUNEL$^+$ cells in the WT and $Bloc1s2^{-/-}$ cortices. Results represent the average number of TUNEL-positive cells per imaging field at ×20 magnification within the cortex. 1.33 ± 0.40 for WT, 1.53 ± 0.54 for $Bloc1s2^{-/-}$. p=0.19, n.s., not significant (Student's t-test); n=6 mice. (**C**) Representative staining from the BrdU/Ki67 cell-cycle exit assay in the WT and $Bloc1s2^{-/-}$ cortices. In the cell-cycle exit assay at E13.5 to E14.5, a BrdU pulse was followed 18 hr later by double labeling of BrdU and Ki67. BrdU$^+$/Ki67$^-$ cells (arrows) were considered to be cells exited from the cell cycle. Pictures showed that the number of cells exited from the cell cycle was decreased in mutant cortex compared with WT. (**D**) Quantification of the proportion of cells leaving the cycle [BrdU$^+$/Ki67$^-$ over total BrdU$^+$] in WT and $Bloc1s2^{-/-}$ cortices. 52.61 ± 3.48 for WT, 33.29 ± 2.95 for $Bloc1s2^{-/-}$. **p<0.01; n=9 mice. (**E**) Immunohistochemistry staining of Tuj1 on sections of WT and $Bloc1s2^{-/-}$ brains at E14.5. The number of Tuj1$^+$ cells was decreased in the $Bloc1s2^{-/-}$ brain. (**F**) Quantification of the thickness of the Tuj1$^+$ neuronal layer at E14.5. 189.70 ± 6.54 for WT, 126.84 ± 10.88 for $Bloc1s2^{-/-}$. ***p<0.001; n=9 mice. (**G**) Immunohistochemistry staining of Pax6 on sections of WT and $Bloc1s2^{-/-}$ brains at E14.5. The number of Pax6$^+$ cells was increased in the $Bloc1s2^{-/-}$ brain. (**H**) Quantification of the number of Pax6$^+$ cells at E14.5. 74.36 ± 3.88 for WT, 113.29 ± 4.95 for $Bloc1s2^{-/-}$. ***p<0.001; n=9 mice. (**I**) Staining of brain sections for BrdU and Ki67 following a 2-hr BrdU pulse at E14.5. Pictures showed an increase in the proportion of cycling cells in S-phase in the $Bloc1s2^{-/-}$ cortex compared with WT. (**J**) Quantification of the BrdU labeling index in WT and $Bloc1s2^{-/-}$ cortices. BrdU labeling index is the percentage of Ki67$^+$ cells that incorporated BrdU. 57.69 ± 1.52 for WT, 70.79 ± 3.27 for $Bloc1s2^{-/-}$; **p<0.01; n=9 mice. (**K**) Immunohistochemistry staining of PH3 on sections of WT and $Bloc1s2^{-/-}$ brains at E13.5 and E14.5. Pictures showed an increase in PH3$^+$ cells in the VZ in the $Bloc1s2^{-/-}$ cortex at E13.5 and E14.5. (**L**) Quantification of the number of PH3$^+$ cells in the VZ at E13.5 and E14.5. 13.31 ± 1.43 for WT at E13.5, 23.49 ± 1.90 for $Bloc1s2^{-/-}$ at E13.5; 4.51 ± 0.25 for WT at E14.5, 7.21 ± 0.66 for $Bloc1s2^{-/-}$ at E14.5. ***p<0.001; n=9 mice per group. All graphs are mean ± s.e.m. Scale bars: 50 μm in **A**, **C**, **E**, **G**, **I** and **K**.

to pregnant dams at E14.5 to label all cells in S phase and measured the ratio of cycling cells that were in S phase in WT and $Bloc1s2^{-/-}$ animals (*Siegenthaler et al., 2009*). Double labeling of BrdU and Ki67 showed an increased labelling index (BrdU$^+$ Ki67$^+$/Ki67$^+$) for cortical progenitors in $Bloc1s2^{-/-}$ mice, suggesting increased proliferation of NPCs in the mutant (*Figure 2I and J*). Furthermore, neural precursor proliferation was also assessed using the mitotic marker phosphohistone-3 (PH3) (*Rash et al., 2011*). When compared with WT, $Bloc1s2^{-/-}$ cortex showed an increased density of PH3$^+$ cells in the VZ at both E13.5 and E14.5 (*Figure 2K and L*). Taken together, NPCs in the $Bloc1s2^{-/-}$ cortex were arrested in a proliferative state, inhibiting normal neurogenesis. Our data demonstrate that BLOS2 is important for the transition from proliferation to differentiation of NPCs.

## BLOS2 ablation leads to elevated Notch signaling in neural progenitor cells

We then investigated the potential signaling pathways involving BLOS2 in the transition from proliferation to differentiation of NPCs in the cortex. We tested the Notch pathway for the following reasons. Notch signaling plays central roles in the coordination of proliferation of NPCs and neuronal differentiation during neurogenesis (*Fortini, 2009*; *Kopan and Ilagan, 2009*). Moreover, overexpression of transcription factors Hes1 and Hes5, which mediate Notch signaling, results in neural progenitor hyperproliferation and impaired neuronal differentiation in the brain (*Ohtsuka et al., 2001*). Similarly, loss of Numb and Numblike in dorsal forebrain leads to the hyperactivation of Notch signaling, increases neural progenitor proliferation, delays cell cycle exit and inhibits neuronal differentiation, mimicking the $Bloc1s2^{-/-}$ phenotypes in NPCs (*Li et al., 2003a*).

To determine whether BLOS2 functions in the control of proliferation and differentiation of NPCs through regulation of Notch signaling, we first examined the production of cleaved Notch (or NICD) using anti-Notch1 V1744, an antibody specific for NICD1. Immunoblotting analyses of cortex extracts derived from $Bloc1s2^{-/-}$ mutants and WT littermates showed that the $Bloc1s2^{-/-}$ cortices contained higher levels of NICD1 than WT animals, indicative of elevated Notch signaling. Likewise, we found an increase in the level of Notch1 transmembrane subunit (N$^{TM}$) in the mutant cortex (*Figure 3A and B*). Immunohistochemical analyses revealed that Notch1 was mainly detected in the ventricular and subventricular areas. In E14.5 $Bloc1s2^{-/-}$ animals, Notch1 intensity was increased compared to the controls (*Figure 3C and D*). The elevated level of Notch1 (N$^{TM}$) in $Bloc1s2^{-/-}$ animals potentially explained their higher level of NICD1. To exclude the possibility of higher expression of Notch ligands in mutant mice, Delta-like 1(Dll1) and Jagged1 (Jag1), two Notch ligands reported to be expressed in neural progenitors, were examined and there was no difference between WT and $Bloc1s2^{-/-}$ cortices (*Figure 3—figure supplement 1A and B*). In addition, we did not find apparent changes in *Itch* and *Dtx2* in $Bloc1s2^{-/-}$ cells, two mammalian homologs of Su(Dx) and Deltex in flies (*Figure 3—figure supplement 1C*), suggesting that the upregulated Notch1 activity is independent of the Deltex-Su(Dx) E3-ubiquitin pathway (*Cornell et al., 1999*).

Next, we assessed the expression levels of Notch target genes *Hes1*, *Hes5* and *Fabp7*. Quantitative reverse transcriptase PCR (qRT-PCR) analyses showed that the levels of *Hes1*, *Hes5*, *Fabp7* mRNA were increased in the $Bloc1s2^{-/-}$ cortex at E14.5 (*Figure 3E–G*). The overall protein levels of Hes1, Hes5 and Fabp7 were likewise increased in the $Bloc1s2^{-/-}$ cortex lysates, indicative of elevated Notch signaling during neurogenesis in mutant cortices (*Figure 3H and I*). We confirmed the effect of BLOS2 on Notch signaling in E14.5 mouse embryonic fibroblasts (MEFs). *Hes1*, *Hes5* and *Fabp7* were downregulated in BLOS2-overexpressed MEFs (*Figure 3—figure supplement 1D–F*) in which Notch1 was reduced and less colocalized with the late endosome marker LBPA (*Figure 3—figure supplement 2A* to *Figure 3—figure supplement 2D*). Furthermore, we observed increased colocalization of Notch1 with LBPA, but decreased colocalization of Notch1 with LAMP1 (a lysosome marker) in cultured NPCs (*Figure 3—figure supplement 2E* to *Figure 3—figure supplement 2H*).

By contrast, the levels of Notch1 (N$^{TM}$), NICD1 and Hes5 were significantly increased in $Bloc1s2^{-/-}$ MEFs. In addition, increased levels of Notch1 (N$^{TM}$) and Notch signaling in $Bloc1s2^{-/-}$ MEFs were mostly rescued by overexpression of full-length FLAG-BLOS2 (*Figure 3—figure supplement 1G and H*). Furthermore, the elevated *Hes1* in $Bloc1s2^{-/-}$ MEFs could be rescued by treatment with a γ-secretase inhibitor, DAPT (*Figure 3—figure supplement 3A*).

To further confirm the effects of Notch elevation on NPC, we found that the average diameter of neurospheres derived from E14.5 NPCs of $Bloc1s2^{-/-}$ mice was significantly increased, a feature that could be rescued by DAPT treatment (*Figure 3—figure supplement 3B*). Likewise,

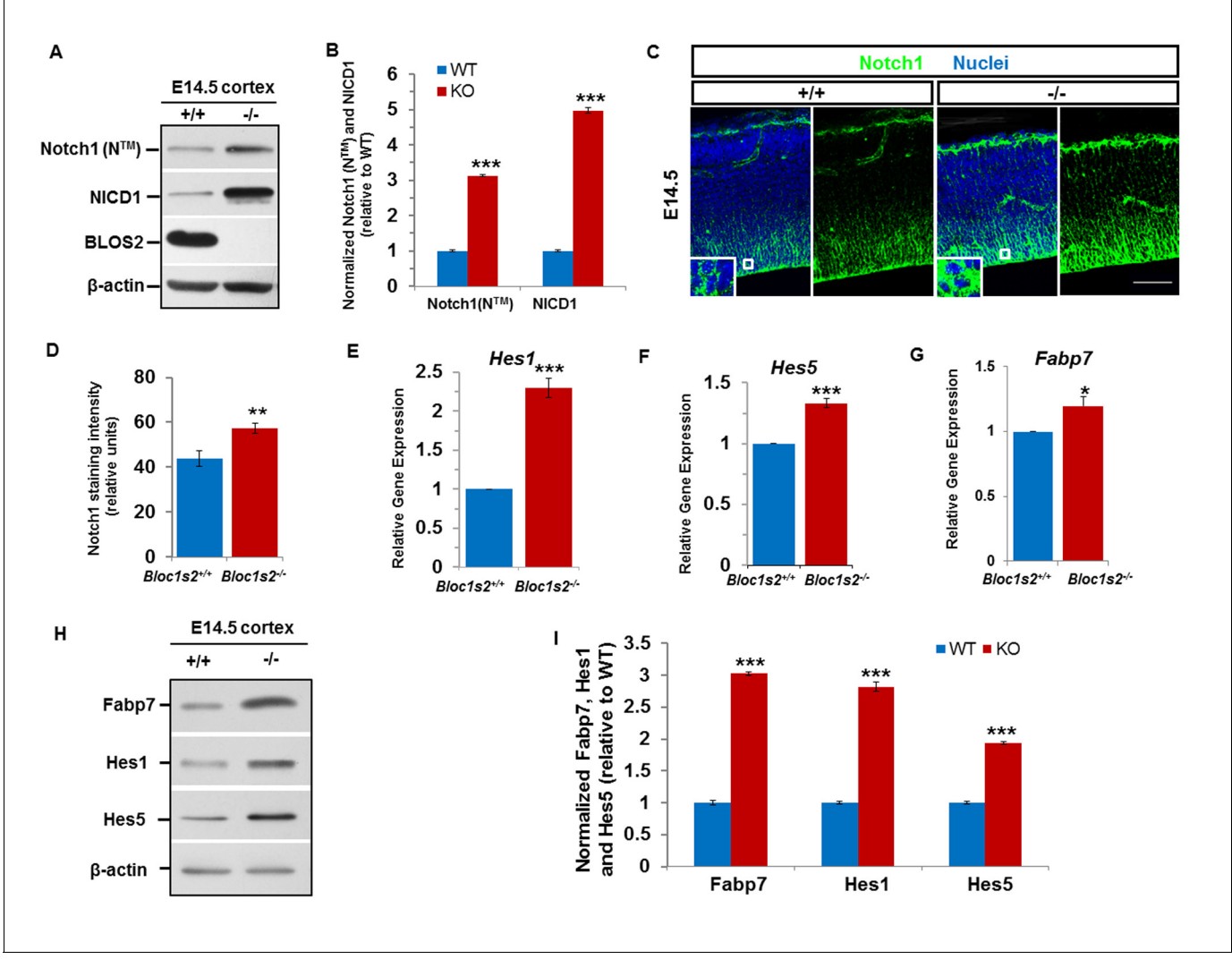

**Figure 3.** Loss of BLOS2 led to increased Notch signaling in mouse brain. (**A**) Immunoblotting analysis of NICD1 and Notch1 (N$^{TM}$) in cortical extracts derived from WT and *Bloc1s2*$^{-/-}$ mice at E14.5 (*Figure 3—source data 1*). The results showed that the levels of NICD1 and Notch1 (N$^{TM}$) were significantly increased in the *Bloc1s2*$^{-/-}$ cortex lysates. (**B**) Quantification of NICD1 and Notch1 (N$^{TM}$) relative to WT cortex. Data are presented as fold change normalized to the mean of WT ± s.e.m. ***$p<0.001$ (Student's t-test); n=9 mice. (**C**) Immunohistochemistry staining of Notch1 on sections of WT and *Bloc1s2*$^{-/-}$ brains at E14.5. Insets are high-magnification images of the boxed regions. Increased Notch1 expression was observed in *Bloc1s2*$^{-/-}$ cortices at E14.5. Scale bar: 50 μm. (**D**) Quantification of signal intensity of Notch1. **$p<0.01$; n=9 mice. (**E-G**) qRT-PCR analysis for *Hes1*, *Hes5* and *Fabp7* of the WT and *Bloc1s2*$^{-/-}$ dissected cortices at E14.5. The expression of *Hes1*, *Hes5* and *Fabp7* were significantly higher in the *Bloc1s2*$^{-/-}$ cortex. Data are presented as fold change normalized to the mean of WT. *$p<0.05$; ***$p<0.001$; n=9 mice. (**H**) Immunoblot analysis of Fabp7, Hes1 and Hes5 in cortical extracts derived from WT and *Bloc1s2*$^{-/-}$ mice at E14.5 *Figure 3—source data 2*). (**I**) Quantification of Fabp7, Hes1 and Hes5 relative to WT cortex. Data are presented as fold change normalized to the mean of WT. ***$p<0.001$; n=9 mice.

The following source data and figure supplements are available for figure 3:

**Source data 1.** Western blots of *Figure 3A*.
**Source data 2.** Western blots of *Figure 3H*.
**Figure supplement 1.** Unchanged Notch ligands in cortices and elevated Notch levels in MEFs.
**Figure supplement 1—source data 1.** Western blots of *Figure 3—figure supplement 1B*.
**Figure supplement 1—source data 2.** Western blots of *Figure 3—figure supplement 1G*.

*Figure 3 continued on next page*

Figure 3 continued

**Figure supplement 2.** Endosomal Notch1 levels are reduced in BLOS2 overexpressed MEFs but increased in BLOS2-KO NPCs.
**Figure supplement 2—source data 1.** Western blots of *Figure 3—figure supplement 2A*.
**Figure supplement 3.** Rescue assays using a γ-secretase inhibitor DAPT.

neuronal differentiation of NPCs from $Bloc1s2^{-/-}$ mice was greatly inhibited, again a feature that could be recovered by DAPT treatment (*Figure 3—figure supplement 3C–E*). These collective data suggest that loss of BLOS2 results in upregulation of Notch signaling in neural progenitors, which inhibits cortical neurogenesis in $Bloc1s2^{-/-}$ mice.

## Loss of BLOS2 in mouse results in increased frequency and number of HSPCs, and elevated Notch signaling in the AGM region

To explore whether embryonic hematopoiesis is impaired as shown in *Figure 1A*, we analyzed the property of HSPCs located in the aorta-gonad-mesonephros (AGM) region of $Bloc1s2^{-/-}$ knockout mice. We confirmed that BLOS2 was null in the AGM of $Bloc1s2^{-/-}$ mice (*Figure 4A*). Cell-sorting analysis showed that $Bloc1s2$ mRNA was expressed in endothelial cells and in HSPCs and non-hematopoietic cells in the AGM, suggesting its ubiquitous expression in hematopoietic lineages and surrounding cells (*Figure 4B*). Strikingly, immunofluorescence analysis showed that expression of Runx1 (a HSPC marker) was increased in the AGM and even expanded into the dorsal region of the dorsal aorta (*Figure 4C*), suggesting that more HSPCs were produced in $Bloc1s2^{-/-}$ mice. The cell proportion of c-Kit$^+$CD34$^+$ HSPCs from E11 AGM in $Bloc1s2^{-/-}$ embryos was higher than that in $Bloc1s2^{+/+}$ embryos (*Figure 4D*), further confirming this notion. The colony-forming unit-cell (CFU-C) assay showed decreased colony formation ability (CFU-Mix) with relatively unchanged CFU-E and drastically decreased CFU-GM from AGM tissues in $Bloc1s2^{-/-}$ mice (*Figure 4E*), suggesting that the differentiation of HSPCs is greatly inhibited. To test whether the increased HSPC production was due to increased cell proliferation, we performed a Ki67-staining assay and the results showed that more Ki67-positive cells were observed in the AGM of $Bloc1s2^{-/-}$ mice (*Figure 4F*). Together, these data suggest that BLOS2 deficiency leads to increased HSPC production in mouse.

Since increased Notch signaling was observed in $Bloc1s2^{-/-}$ brains, we wondered whether this also holds true in the AGM region. As expected, the protein level of both Notch1 and NICD in the AGM region was increased in $Bloc1s2^{-/-}$ mice (*Figure 4G*). FACS analysis with purified CD31$^+$ cells further confirmed this result (*Figure 4H*). We found increased colocalization of Notch1 with LBPA and decreased colocalization of Notch1 with LAMP1 in the AGM region (*Figure 4—figure supplement 1A* to *Figure 4—figure supplement 1D*). Next, we examined the expression of several Notch target genes. qRT-PCR analysis showed that the expression of *Hey1* and *Hey2* was significantly upregulated, suggesting that Notch signaling is enhanced (*Figure 4I*).

## BLOS2 regulates HSPC development through inhibition of Notch activity in zebrafish

To determine whether BLOS2 is evolutionarily conserved in HSPC development in vertebrates, we first examined the expression of *bloc1s2* in zebrafish. We found that *bloc1s2* was expressed in neurons and vessels (*Figure 5—figure supplement 1A*), especially in endothelial cells (*Figure 5—figure supplement 1B*), which indicates a potential role of *bloc1s2* in zebrafish hematopoiesis. To study its function in zebrafish, *bloc1s2* mutants were generated by CRISPR/Cas9 systems. The gRNA was designed to target 3 of the 4 *bloc1s2* transcripts (*Figure 5—figure supplement 2A*), and the expression of *bloc1s2* was significantly decreased, but not abolished because of the expression of transcript *bloc1s2-201* in the *bloc1s2* mutant fish, which we designated as a hypomorphic mutant, *bloc1s2 hypo* (*Figure 5A* to *Figure 5C*; *Figure 5—figure supplement 2B* to *Figure 5—figure supplement 2D*). To explore the function of *bloc1s2* in zebrafish hematopoiesis, we examined the expression of HSPC markers in the AGM region of *bloc1s2* mutants. Strikingly, we found that the

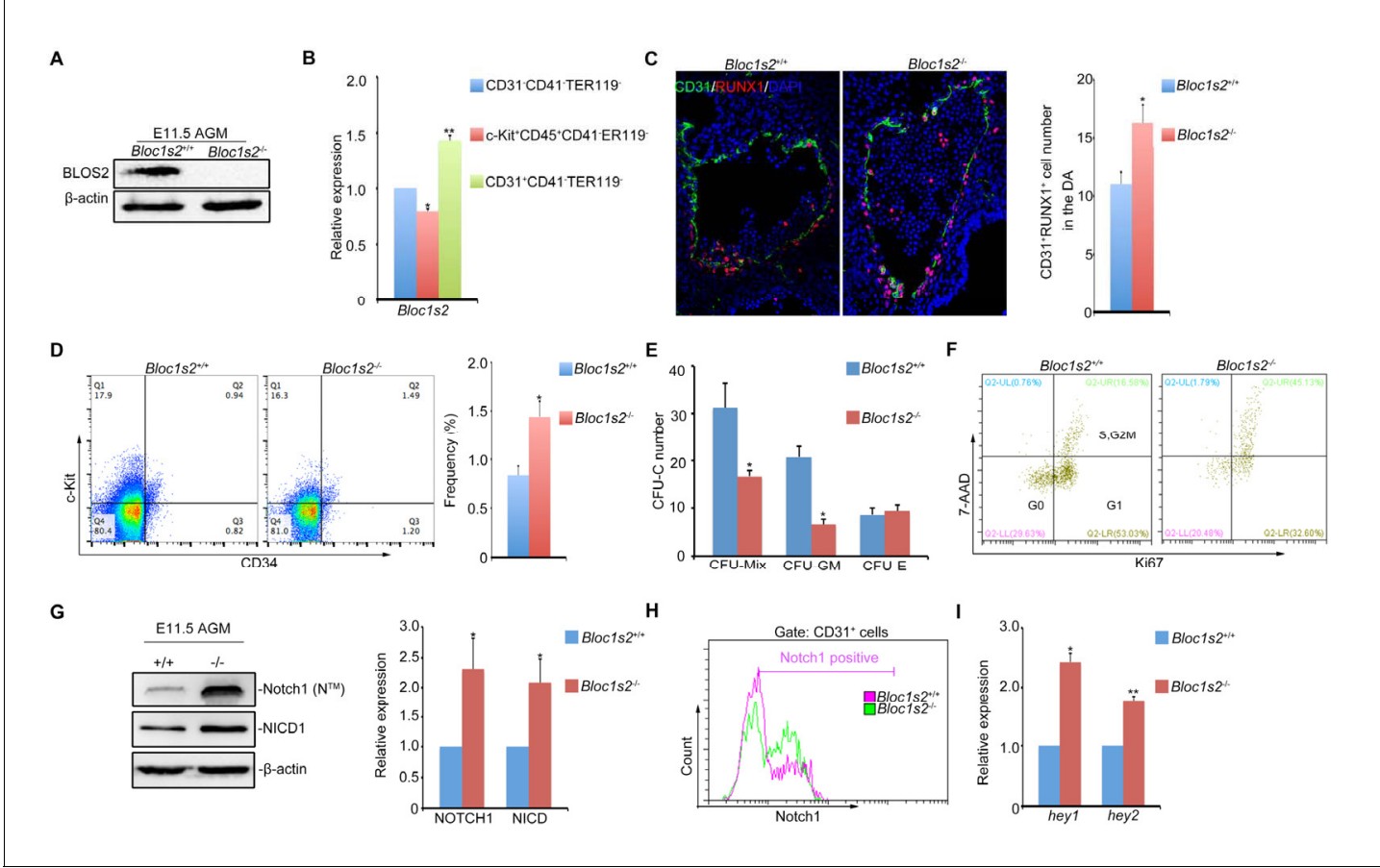

**Figure 4.** Loss of BLOS2 drives HSPCs from quiescence to rapid proliferation and results in increased frequency and number of HSPCs and elevated Notch signaling. (**A**) Immunoblotting of BLOS2 in the E11.5 AGM region of $Bloc1s2^{+/+}$ and $Bloc1s2^{-/-}$ embryos (***Figure 4—source data 1***). The cells in the AGM region were immunoblotted with anti-BLOS2 and anti-β-actin. (**B**) qRT-PCR results showed that $Bloc1s2$ was enriched in $CD31^+CD41^-CD45^-TER119^-$ endothelial cells. Each bar represents the mean ± s.e.m of three independent samples. *p<0.05; **p<0.01 (Student's t-test). (**C**) The immunofluorescence of Runx1 and CD31 expression in the E11 AGM region of $Bloc1s2^{+/+}$ and $Bloc1s2^{-/-}$ embryos. Quantification of $CD31^+$ Runx1$^+$ cell numbers in the DA (dorsal aorta). *p<0.05 (n=3 mice). (**D**) The cell proportion of c-Kit$^+$CD34$^+$ HSPC from E11 AGM in $Bloc1s2^{+/+}$ and $Bloc1s2^{-/-}$ embryos. *p<0.05 (n=3 mice). (**E**) Each well plated with 600 c-Kit$^+$CD34$^+$ HSPCs from E11 $Bloc1s2^{+/+}$ and $Bloc1s2^{-/-}$ embryos was used for CFU-C assay. Seven days later, the numbers of colonies were counted. Each bar represents the mean ± s.e.m of three independent samples. CFU-MIX: 15.67 ± 2.51 for WT, 8.33 ± 0.58 for $Bloc1s2^{-/-}$ (*p<0.05); CFU-E: 4.33 ± 0.71 for WT, 4.67 ± 0.65 for $Bloc1s2^{-/-}$ (p=0.423); CFU-GM: 10.33 ± 1.63 for WT, 3.33 ± 0.545 for $Bloc1s2^{-/-}$ (p<0.05). (**F**) $Bloc1s2$-deficient c-Kit$^+$CD34$^+$ HSPCs have increased proliferation, indicated as Ki67$^+$ 7AAD$^+$ (7-amino-actinomycin D). (**G**) Immunoblotting of Notch signaling in the AGM region. The cells in the AGM region were immunoblotted with anti-Notch1, anti-NICD and anti-β-actin (***Figure 4—source data 2***). The right panel shows a quantitative analysis of the western blotting results. Each bar represents the mean ± s.e.m of three independent samples. *p<0.05, **p<0.01. (**H**) Fluorescence-activated cell sorting (FACS) analysis of Notch1 expression on $CD31^+CD41^-CD45^-TER119^-$ endothelial cells in $Bloc1s2^{+/+}$ or $Bloc1s2^{-/-}$ embryos. (**I**) qPCR analysis of $hey1$ and $hey2$ expression in $CD31^+CD41^-CD45^-TER119^-$ endothelial cells from $Bloc1s2^{+/+}$ or $Bloc1s2^{-/-}$ embryos. Each bar represents the mean ± s.e.m of three independent samples. *p<0.05, **p<0.01.

The following source data and figure supplement are available for figure 4:

**Source data 1.** Western blots of *Figure 4A*.

**Source data 2.** Western blots of *Figure 4G*.

**Figure supplement 1.** Endosomal Notch1 levels are increased in BLOS2-KO AGM cells.

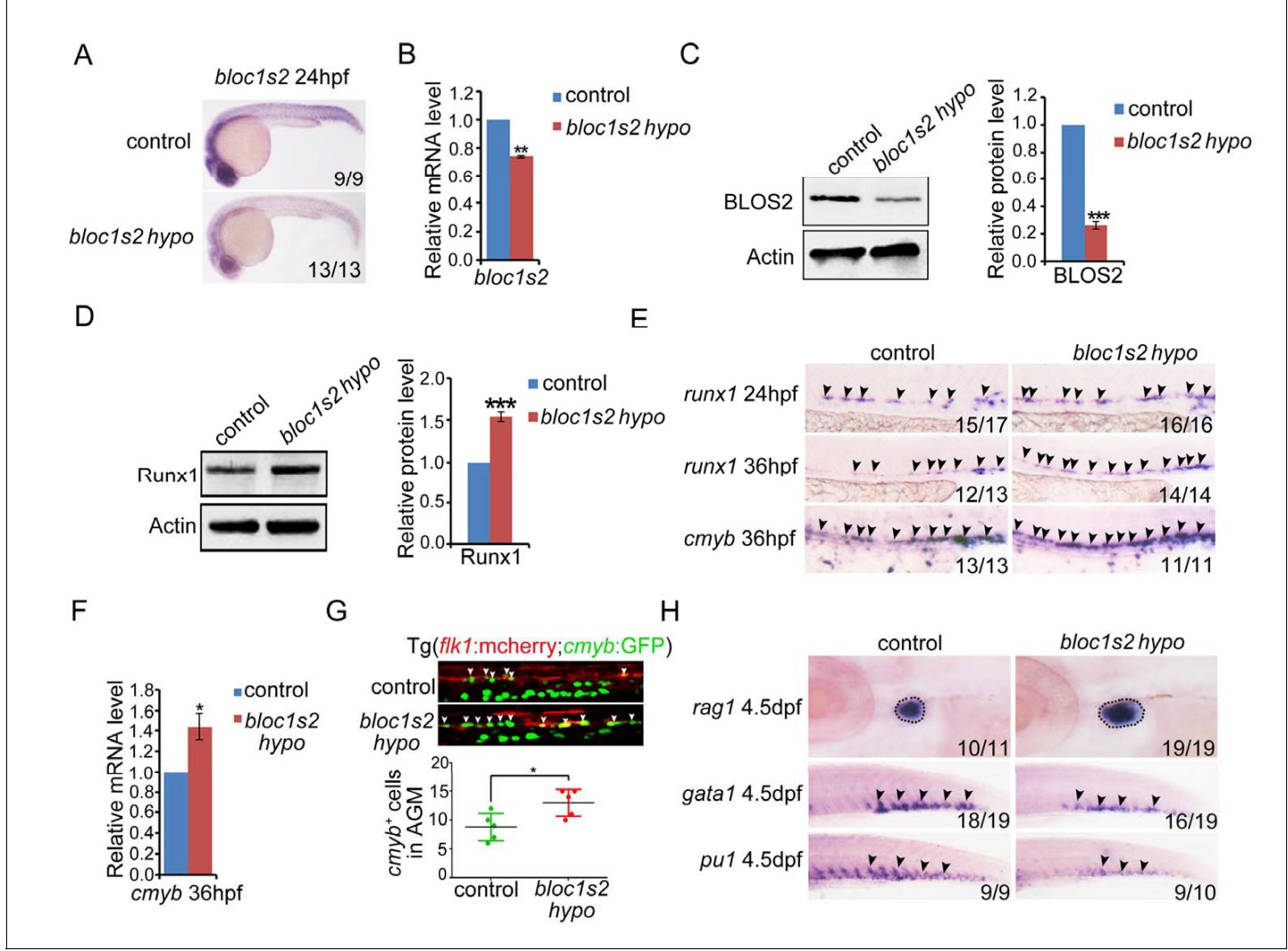

**Figure 5.** *bloc1s2* deficiency promotes HSPC production in zebrafish. (**A-C**) The results of (**A**) whole mount in situ hybridization (WISH), (**B**) qRT-PCR and (**C**) immunoblotting (***Figure 5—source data 1***) showed that the expression of *bloc1s2* at the RNA and protein levels in *bloc1s2* mutants was decreased but not abolished compared to that in control embryos at 24 hpf. Each bar represents the mean ± s.e.m of three independent experiments. **p<0.01, ***p<0.001 (Student's t-test). (**D**) The Runx1 protein level was increased in *bloc1s2* mutants (***Figure 5—source data 2***). The right panel is the quantitative analysis of the western blotting results. Each bar represents the mean ± s.e.m of three independent experiments. ***p<0.001. (**E**) The WISH results showed that the expression of *runx1* at 24 hpf and 36 hpf, and *cmyb* expression at 36 hpf were increased in *bloc1s2* mutants. Black arrowheads mark expression of *runx1* and *cmyb* in the AGM region. (**F**) qRT-PCR results from the dissected trunk region showed that expression of *cmyb* at 36 hpf was significantly increased in *bloc1s2* mutants. Each bar represents the mean ± s.e.m of three independent experiments. *p<0.05. (**G**) We generated the *bloc1s2* mutants in a Tg (*flk1*:mcherry;*cmyb*:GFP) background. The upper panels show that the number of *flk1*⁺*cmyb*⁺ cells in outcrossed embryos was increased. White arrowheads mark *flk1*⁺*cmyb*⁺ cells in the AGM region at 36 hpf. The lower panel is the quantitative analysis of the western blotting results. Each bar represents the mean ± s.e.m of seven embryos. **p<0.01. (**H**) *rag1* expression was increased in the thymus region, but *gata1* and *pu1* expression was decreased in the caudal hematopoietic tissue (CHT) region in *bloc1s2* mutants at 4.5 dpf. Black dashed circles mark *rag1* expression in the thymus region. Black arrowheads mark expression of *gata1* and *pu1* in the CHT region.

The following source data and figure supplements are available for figure 5:

**Source data 1.** Western blots of ***Figure 5C***.
**Source data 2.** Western blots of ***Figure 5D***.
**Figure supplement 1.** BLOS2 regulation of HSPC development in zebrafish is gene-specific.
**Figure supplement 2.** Genome editing of zebrafish *bloc1s2* by Cas9 technology.
*Figure 5 continued on next page*

*Figure 5 continued*

**Figure supplement 2—source data 1.** Western blots of *Figure 5—figure supplement 2D*.

**Figure supplement 3.** Increased Notch activity mediates the HSPC phenotype in zebrafish or mouse *bloc1s2* mutants.

**Figure supplement 3—source data 1.** Western blots of *Figure 5—figure supplement 3C*.

protein level of Runx1 was increased in *bloc1s2* mutants (*Figure 5D*). In addition, the mRNA expression of *runx1* and *cmyb* (another HSPC marker) was also increased in the AGM region (*Figure 5E and F*). To further confirm this HSPC phenotype, *bloc1s2* homozygous mutants were outcrossed with the Tg(*flk1*:mcherry;*cmyb*:GFP) transgenic line, and finally we obtained the *bloc1s2* homozygotes in the Tg(*flk1*:mcherry;*cmyb*:GFP) background. Confocal imaging showed an increased number of hemogenic endothelium (HE) and emerging HSPCs in the *bloc1s2* mutants (*Figure 5G*). Moreover, the increased expression of HSPC markers (*runx1* and *cmyb*) in *bloc1s2* mutants was rescued by overexpression of *bloc1s2* itself (*Figure 5—figure supplement 1C and D*), indicating the specific function of *bloc1s2* in HSPC development.

The emerging HSPCs are derived from specified HE through a process named endothelial-to-hematopoietic transition (EHT). Thus, the increased HSPCs may be a result of a high proliferation ratio of HE or emerging HSPCs. To test this hypothesis, BrdU and PH3 assays were performed. As expected, the BrdU signals around the dorsal aorta were significantly increased (*Figure 5—figure supplement 1E*). To investigate whether the increased HSPCs in *bloc1s2* mutants can differentiate into blood cells normally, the expression of lymphoid (*rag1*), erythroid (*gata1*) and myeloid (*pu1*) markers was examined. We found that lymphoid differentiation was increased, while erythroid and myeloid differentiation was impaired (*Figure 5H*). Taken together, these data suggest that ablation of BLOS2 promotes HSPC generation, and also facilitates lymphoid differentiation at the expense of erythroid and myeloid differentiation.

We next explored whether BLOS2 regulates HSPC generation, as it regulates NPC production, through the Notch signaling pathway . We first determined the expression of Notch targets in *bloc1s2* mutants, showing that *hey2* and *ephrinB2a* expression was increased (*Figure 5—figure supplement 3A and B*). Notch1 protein level was also increased in *bloc1s2* mutants (*Figure 5—figure supplement 3C*). Moreover, we obtained the *bloc1s2* homozygotes in a Notch reporter line in the Tg(*fli1a*:EGFP;*tp1*:dsRed) background. We found that the $tp1^+$ signals in the dorsal aorta were increased in the *bloc1s2* mutants (*Figure 5—figure supplement 3D*). To further confirm whether the increase in Notch activity mediated the increase of HSPC generation in *bloc1s2* mutants, a γ-secretase inhibitor DBZ was used to perform the rescue experiments. As expected, the increased expression of Notch targets *hey2* and *ephrinB2a*: HSPC markers *runx1* and *cmyb* in *bloc1s2* mutants was blocked by the DBZ treatment (*Figure 5—figure supplement 3E and F*). Consistent with the zebrafish data, DBZ treatment inhibited the expansion of c-Kit⁺CD34⁺ HSPCs from E11 AGM in $Bloc1s2^{-/-}$ mouse embryos (*Figure 5—figure supplement 3G*). Taken together, these results suggest that elevated Notch signaling leads to increased HSPC generation in zebrafish and mice, and that the conservation of BLOS2-regulated Notch signaling is an important axis during HSPC development in vertebrates.

## BLOS2 is required for lysosomal degradation of Notch1

To further investigate the underlying mechanism utilized by BLOS2 in regulating the protein level of Notch1 ($N^{TM}$), we first investigated the expression of Notch1 in the WT and $Bloc1s2^{-/-}$ cortices. qRT-PCR analysis showed that there was no significant change in *Notch1* mRNA level between WT and $Bloc1s2^{-/-}$ cortices at E14.5 (*Figure 6A*), suggesting that the increase of Notch1 ($N^{TM}$) in the mutant cortex does not arise from the increased transcription of Notch1.

Intracellular Notch1 ($N^{TM}$) is targeted either to the proteosomal or to the endosomal/lysosomal pathway for protein degradation (*Andersen et al., 2012*). To determine whether proteasomal or lysosomal degradation of Notch1 is impaired in $Bloc1s2^{-/-}$ MEFs, we determined the response of

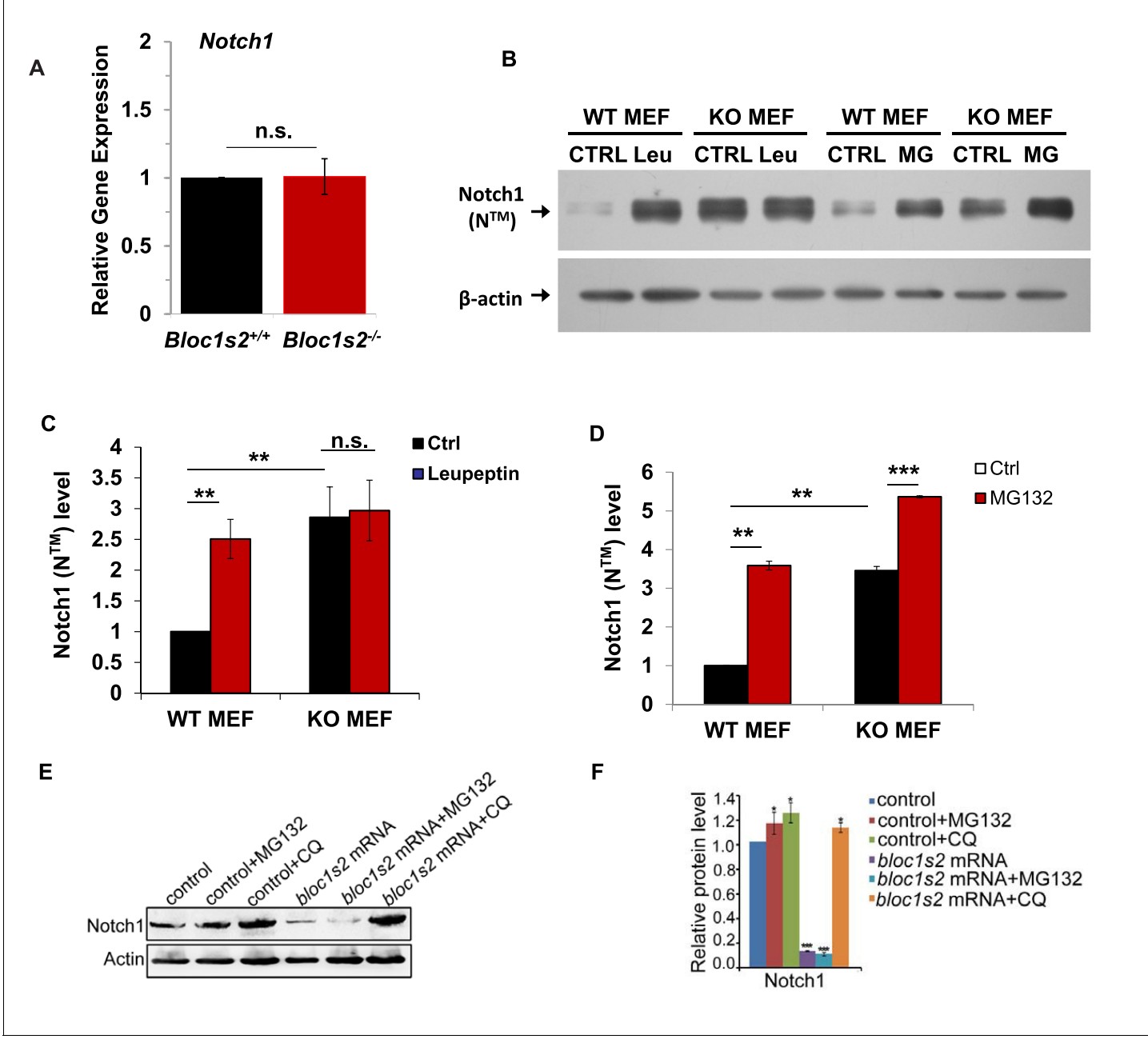

**Figure 6.** Lysosomal degradation of Notch1 was impaired in mouse and zebrafish *Bloc1s2*[−/−] cells. (**A**) qRT-PCR analysis for *Notch1* of the WT and *Bloc1s2*[−/−] dissected cortices at E14.5. No change in *Notch1* mRNA levels between WT and *Bloc1s2*[−/−] cortices was observed. Data are presented as fold change normalized to the mean of WT. n.s., not significant, p=0.94 (Student' s t-test; n=9 mice). (**B**) WT and *Bloc1s2*[−/−] MEFs incubated with or without leupeptin (100 µM) or MG132 (10 µM) for 4 hr at 37°C and immunoblotted for Notch1 (N[TM]) (*Figure 6—source data 1*). (**C** and **D**) Quantification of Notch1 (N[TM]) relative to that in untreated WT MEFs. **p<0.01; ***p<0.001; n.s., not significant, p=0.876 (n=6). All graphs are mean ± s.e.m. (**E**) Control and zebrafish *bloc1s2* mRNA overexpressing embryos were treated with 50 µM chloroquine from 18 hpf to 36 hpf. The immunoblotting results (*Figure 6—source data 2*) showed that the decreased Notch1 protein level in *bloc1s2* overexpressing embryos was recovered by chloroquine treatment, but not changed by MG132 treatment. (**F**) Quantitative analysis of the immunoblotting results. Each bar represents the mean ± s.e.m of three independent experiments. *p<0.05; ***p<0.001.

The following source data is available for figure 6:

**Source data 1.** Western blots of *Figure 6B*.
**Source data 2.** Western blots of *Figure 6E*.

Notch1 (N$^{TM}$) to lysosome or proteasome inhibition. Upon treatment of WT MEFs with leupeptin (an inhibitor of lysosomal proteolysis) or MG132 (a proteasome inhibitor), we found that Notch1 (N$^{TM}$) degradation was suppressed by either leupeptin or MG132, indicating that a portion of Notch1 (N$^{TM}$) is degraded in lysosomes and part of Notch1 (N$^{TM}$) is degraded by the proteasome (*Figure 6B*). In *Bloc1s2$^{-/-}$* MEFs, we observed no effect of leupeptin treatment on Notch1 (N$^{TM}$) levels, suggesting that the increased Notch1 (N$^{TM}$) shown in *Bloc1s2$^{-/-}$* MEFs is probably due to reduced Notch1 (N$^{TM}$) degradation by lysosomes (*Figure 6C and D*). However, Notch1 (N$^{TM}$) degradation was suppressed by MG132 in *Bloc1s2$^{-/-}$* MEFs, similar to the effect in WT MEFs, implying that the proteasomal pathway of Notch1 (N$^{TM}$) degradation is not affected by loss of BLOS2 (*Figure 6C and D*). These results suggest that BLOS2 is probably required for lysosomal degradation of Notch1 (N$^{TM}$).

In order to find out whether the inhibition of Notch activity in zebrafish *bloc1s2$^{-/-}$* also resulted from impaired lysosomal degradation, we used the lysosomal inhibitor chloroquine to treat the *bloc1s2* mRNA overexpressing embryos. It showed that the decreased Notch1 expression in *bloc1s2* overexpressing embryos was restored by the chloroquine treatment, but not by treatment with the proteasomal inhibitor MG132 (*Figure 6E and F*), supporting the hypothesis that BLOS2 inhibits Notch signaling pathway mainly through lysosome degradation. Altogether, these results demonstrated that BLOS2 regulates NPC or HSPC development through inhibition of Notch activity in a lysosome-dependent manner.

## BLOS2 interacts with NICD during lysosomal trafficking of Notch1

The fact that BLOS2 is a shared subunit of BLOC-1 and BORC, which has been reported to function in endo-lysosomal trafficking, suggests the possible impairment of Notch1 in endo-lysosomal trafficking in association with loss of BLOS2. We first performed a 5–20% (w/v) OptiPrep gradient separation of the homogenates of WT and *Bloc1s2$^{-/-}$* brains (*Zhang et al., 2014b*). Using organellar markers, EEA1 (early endosomes), CD63 (MVBs and late endosomes), LAMP1 (lysosomes), and GM130 (Golgi), we observed that the localization of Notch1 (N$^{TM}$) exhibited a marked increase in the EEA1 and CD63-labeled fractions in *Bloc1s2$^{-/-}$* brains compared to WT, indicative of accumulation of Notch1 (N$^{TM}$) in endosomes with loss of BLOS2 (*Figure 7A and B*). We further examined Notch1 receptor localization by co-labeling cells with Notch1 and markers of endosomes and lysosomes (EEA1, LBPA and LAMP1) (*Delevoye et al., 2009*; *Tomas et al., 2015*). Compared to WT cells, we observed slightly increased endogenous Notch1 localization in EEA1-labeled early endosomes (*Figure 7C and F*) and significantly increased localization in LBPA-labeled MVBs and late endosomes in *Bloc1s2$^{-/-}$* MEFs (*Figure 7D and G*). Using structured illumination microscopy (SIM), we confirmed the accumulation of Notch1 receptor in LBPA-positive vesicles in *Bloc1s2$^{-/-}$* MEFs (*Figure 7—figure supplement 1A*). However, we did not observe apparent accumulation of Notch1 in *Bloc1s2$^{-/-}$* MEFs labeled with internalized TfA568 after treatment for 30 min representing recycling endosomes (*Figure 7—figure supplement 1B*) (*Delevoye et al., 2016*). By contrast, loss of BLOS2 resulted in a significantly decreased Notch1 localization in LAMP1-labeled lysosomes (*Figure 7E and H*). These results suggested that the endo-lysosomal trafficking of Notch1 receptor is altered in *Bloc1s2$^{-/-}$* MEFs, and that the accumulated Notch1 in the endosomes promotes its signaling.

Finally, we evaluated whether BLOS2 physically interacts with Notch1 to mediate its lysosomal targeting. In our co-immunoprecipitation (Co-IP) assays, we found that BLOS2 bound to the Notch1 intracellular domain (NICD1) in HEK293T cells expressing 3×Flag-NICD1 and Myc-BLOS2 (*Figure 8A*). By endogenous immunoprecipitation, BLOS2 was precipitated by anti-Notch1 with cells dissected from mouse E11 AGM tissues (*Figure 8B*), suggesting that BLOS2 and Notch1 do indeed interact with each other. We also found that the fusion protein Flag-BLOS2 signal colocalized with the signal of 6×Myc-Notch1 in HeLa cells (*Figure 8C*), which partially co-stained with LysoTracker (*Figure 8—figure supplement 1A*). These results suggest that BLOS2 mediates the endo-lysosomal trafficking of Notch1 probably via its interaction with the NICD1 region.

BLOS2 is a shared subunit of BLOC-1 and BORC (*Langemeyer and Ungermann, 2015*; *Pu et al., 2015*). It is not clear whether BLOC-1 functions as a universal mechanism in mediating the endo-lysosomal trafficking of different signaling receptors, or whether different BLOC-1 subunits play specific roles in mediating different signaling receptors. We first investigated whether Notch1 levels were altered in mutants with loss of other BLOC-1 or BORC subunits. We chose mouse mutants of *Pldn* $^{pa}$

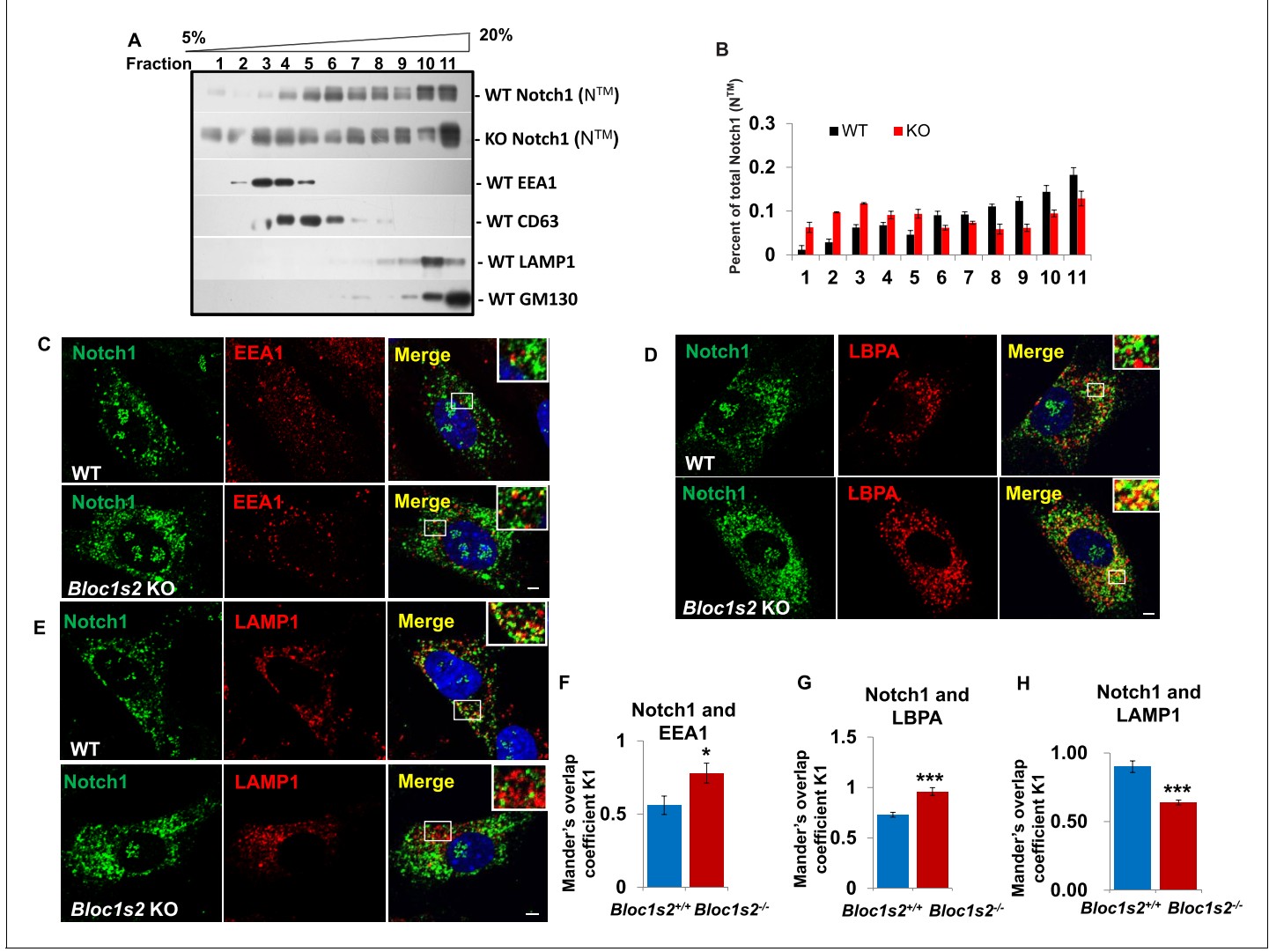

**Figure 7.** The endolysosomal trafficking of Notch1 was altered in *Bloc1s2*⁻/⁻ MEFs. (**A**) OptiPrep gradient fractionation of WT and *Bloc1s2*⁻/⁻ brains at E14.5 probed for Notch1 (N^TM), BLOS2, EEA1 (endosome marker), CD63 (MVBs and late endosomes marker), LAMP1 (lysosome marker), and GM130 (Golgi marker) (***Figure 7—source data 1***). Notch (N^TM) distribution exhibited a marked increase in the EEA1- and CD63-labeled fractions in *Bloc1s2*⁻/⁻ brain tissues. (**B**) Quantification of the Notch1 (N^TM) in (**A**) displayed as the proportion in each fraction relative to the total fractions. (**C** to **E**) WT and *Bloc1s2*⁻/⁻ MEFs immunostained with antibodies against endogenous Notch1 (green) and EEA1/LBPA/LAMP1 (red). This imaging revealed increased endogenous Notch1 localization in EEA1- and LBPA-positive vesicles, and decreased Notch1 localization in LAMP1-labeled vesicles in *Bloc1s2*⁻/⁻ MEFs. Scale bars: 5 μm. (**F** to **H**) Quantification of Mander's overlap coefficient K1 representing co-localizations of Notch1 with EEA1 and LBPA and LAMP1, respectively. *p<0.05, ***p<0.001 (Student's t-test; n=100 cells). All graphs are mean ± s.e.m.

The following source data and figure supplement are available for figure 7:

**Source data 1.** Western blots of *Figure 7A*.

**Figure supplement 1.** Subcellular localization of Notch1 in late endosomes and recycling endosomes.

(***Huang et al., 1999***) or *Dtnbp1* ^sdy (***Li et al., 2003b***) to represent BLOC-1 only, *Kxd1*⁻/⁻ (***Yang et al., 2012***) to represent BORC only, and *Bloc1s1*⁻/⁻ (***Zhang et al., 2014a***) to represent shared BLOC-1 and BORC. We observed the destabilization of two BLOC-1 subunits (BLOS1 and dysbindin) and drastic reduction of KXD1 in *Bloc1s2*⁻/⁻ MEFs, suggesting its tight associations with BLOC-1 and BORC subunits (***Figure 8—figure supplement 2A***). However, we did not observe apparent changes of Notch1 levels in MEFs isolated from the *Pldn* ^pa, *Kxd1*⁻/⁻, or *Bloc1s1*⁻/⁻ mice

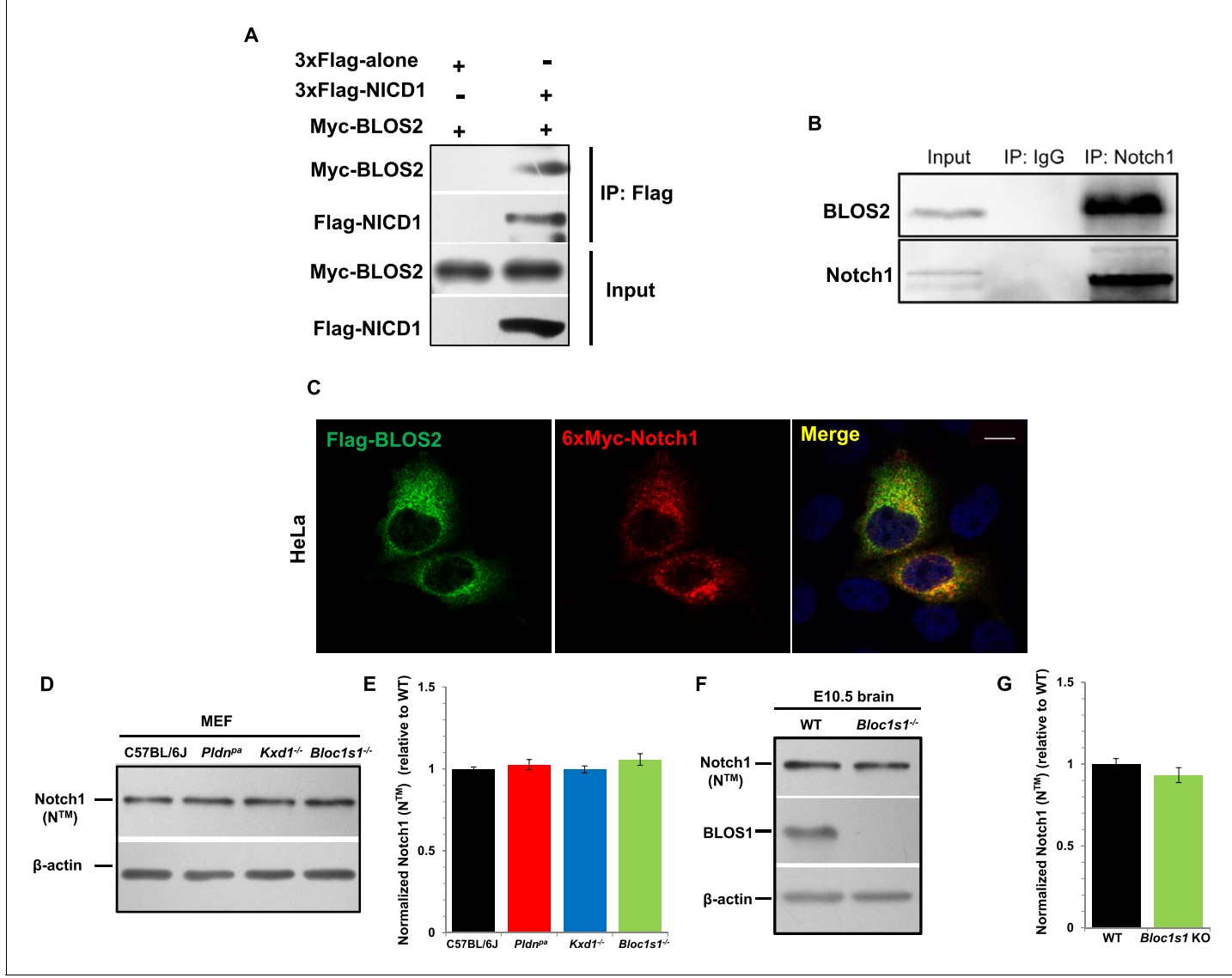

**Figure 8.** BLOS2 physically interacts with NICD1 and functions independently of BLOC-1 and BORC. (**A**) Immunoblotting analysis of Co-IP performed in HEK293T cells. Flag-NICD1 co-eluted with Myc-BLOS2, but Flag-alone did not (*Figure 8—source data 1*). (**B**) Endogenous co-immunoprecipitation of Notch1 with BLOS2 in the AGM region (*Figure 8—source data 2*). IP, immunoprecipitation. (**C**) Partial colocalization of BLOS2 and Notch1 in HeLa cells transfected with the Flag-BLOS2 and 6×Myc-Notch1 constructs. Scale bar: 5 μm. (**D**) Immunoblotting analysis of Notch1 (N$^{TM}$) in WT, *Pldn$^{pa}$*, *Kxd1*-KO, and *Bloc1s1*-KO MEFs (*Figure 8—source data 3*). No changes in Notch1 (N$^{TM}$) levels in *Pldn$^{pa}$*, *Kxd1*-KO, and *Bloc1s1*-KO MEFs were observed. (**E**) Quantification of levels of Notch1 (N$^{TM}$) relative to WT MEFs. Data are presented as fold change normalized to the mean of WT. p=0.457 (*Pldn$^{pa}$*), p=0.901 (*Kxd1*-KO), p=0.178 (*Bloc1s1*-KO) (Student's t-test; n=9). (**F**) Immunoblotting analysis of Notch1 (N$^{TM}$) in brain extracts derived from WT and *Bloc1s1$^{-/-}$* mice at E10.5 (*Figure 8—source data 4*). No change in Notch1 (N$^{TM}$) levels in the *Bloc1s1$^{-/-}$* cortex lysates was observed. (**G**) Quantification of Notch1 (N$^{TM}$) relative to WT cortex. Data are presented as fold change normalized to the mean of WT. p=0.264 (Student's t-test; n=9).

The following source data and figure supplements are available for figure 8:

**Source data 1.** Western blots of *Figure 8A*.

**Source data 2.** Western blots of *Figure 8B*.

**Source data 3.** Western blots of *Figure 8D*.

**Source data 4.** Western blots of *Figure 8F*.

**Figure supplement 1.** Unaltered lysosomal functions and EGFR steady-state levels in *Bloc1s2$^{-/-}$* MEFs or tissues.

*Figure 8 continued on next page*

*Figure 8 continued*

**Figure supplement 1—source data 1.** Western blots of *Figure 8—figure supplement 1B*.

**Figure supplement 1—source data 2.** Western blots of *Figure 8—figure supplement 1D*.

**Figure supplement 1—source data 3.** Western blots of *Figure 8—figure supplement 1F*.

**Figure supplement 2.** Destabilization of other BLOC-1 or BORC subunits in *Bloc1s2⁻/⁻* MEFs and schematic model of lysosomal degradation of Notch1 mediated by BLOS2.

**Figure supplement 2—source data 1.** Western blots of *Figure 8—figure supplement 2A*.

**Figure supplement 3.** Ultrastructures of endo-lysosomal organelles.

(*Figure 8D and E*). Both BLOS1 and BLOS2 are shared subunits of BORC and BLOC-1, and knockout mutants for either of the genes that encode these subunits are embryonic lethal. However, we did not observe apparent change of Notch1 level in the $Bloc1s1^{-/-}$ embryonic brain (*Figure 8F and G*).

Next we examined whether other endocytosed receptors were also altered in $Bloc1s1^{-/-}$ mutant cells. We did not find apparent changes of EGFR levels in either $Bloc1s1^{-/-}$ MEFs (*Figure 8—figure supplement 1B*) or $Bloc1s1^{-/-}$ cortices (*Figure 8—figure supplement 1D*). Furthermore, the lysosomal number and function in $Bloc1s1^{-/-}$ MEFs were apparently normal (*Figure 8—figure supplement 1F*; *Figure 8—figure supplement 3*). These results suggest that the mechanism utilized by BLOS2 specifically regulates Notch1, but is not a general mechanism for lysosomal degradation of other endocytosed receptors, such as EGFR. Taken together, these results suggest that BLOS2 may function as a specific regulator of Notch signaling, which is probably independent of BLOC-1 or BORC.

## Discussion

In this study, we showed that a novel player, BLOS2, is involved in endo-lysosomal trafficking of Notch1 by acting as a negative regulator of Notch signaling. In WT cells, Notch1 receptor is endocytosed to early endosomes, where the receptor is either trafficked back to the membrane by recycling endosomes or directed to lysosomal degradation through MVBs and late endosomes. In mutant cells, loss of BLOS2 resulted in impaired endo-lysosomal trafficking of Notch1, which consequently caused the accumulation of a large fraction of Notch1 in the MVBs and late endosomes. The hyperactivation of Notch signaling led to the defective embryonic corticogenesis and hematopoiesis in the $Bloc1s2^{-/-}$ mutants. We also observed interaction between BLOS2 and Notch1, suggesting that BLOS2 most probably mediates endo-lysosomal trafficking of Notch1 (a model in *Figure 8—figure supplement 2C*). Lysosomal degradation of Notch receptors prevents ectopic Notch activation by turning off the signals. Our observation that BLOS2 functions as a negative regulator of Notch signaling provides new mechanistic insights into the fine-tuning of Notch signaling by endo-lysosomal trafficking.

Control of the proliferation and differentiation of NPCs must be precisely regulated by extrinsic and intracellular signals, so that cortical neurons are organized into specific layers. At developmental stages, these processes are extraordinarily sensitive to the dosage of Notch signaling. We showed here that loss of BLOS2 increased the number of NPCs, delayed cell cycle exit, inhibited neuronal differentiation and consequently led to morphological defects of the cortex. Cortical developmental defects in $Bloc1s2^{-/-}$ mutants phenocopy Emx1$^{IREScre}$-mediated *numb* and *numblike* double mutants, leading us to investigate whether BLOS2 is implicated in the negative regulation of Notch signaling (*Li et al., 2003a*). Indeed, our results showed that loss of BLOS2 resulted in the upregulation of Notch signaling in embryonic cortex, which led to the developmental arrest of NPCs from proliferation to differentiation. Our findings reveal that BLOS2 is a new player involved in the control of cortical progenitor self-renewal and generation of cortical neurons.

Notch signaling is remarkably pleiotropic and could be important for the development of almost all tissues in metazoans (*Louvi and Artavanis-Tsakonas, 2006*). We have observed increased HSPC production in both mouse and zebrafish embryos upon the upregulation of Notch signaling resulting from the loss-of-function of BLOS2. More profound effects could also be explained under this mechanistic controlling of Notch signaling. For example, previous studies have shown that Notch signaling plays important roles in the development of the neural crest and the retinal pigment epithelium (RPE) (*Bao and Cepko, 1997*; *Lee et al., 2005*). Abnormal eye pigmentation and craniofacial malformation in Bloc1s2$^{-/-}$ mice, as described in *Figure 1* could well be attributable to the disruption of Notch signaling during neural crest development. However, loss of eye pigment could also result from disrupted melanosomal biogenesis as BLOS2 may play a role in BLOC-1 in melanosomal protein trafficking. The detailed mechanisms for these defects await further investigation. Nonetheless, the *Bloc1s2*$^{-/-}$ mutant provides a new tool to study the developmental regulation of Notch signaling in embryogenesis and organogenesis.

The cellular mechanisms underlying the control of the strength of Notch signaling are important for biological processes (*Hori et al., 2012*). Recent studies have shown that endo-lysosomal trafficking of the Notch receptor plays important roles in the regulation of Notch signaling (*Fortini, 2009*; *Yamamoto et al., 2010*). Endocytic trafficking has been considered to play a negative role in signaling pathways by transporting Notch receptor to lysosomes for degradation. Also, endocytic trafficking is important for the activation of Notch signaling, which may occur not only on the cell membrane but also in endocytosed vesicles or endosomes (*Hori et al., 2011*; *Sorkin and von Zastrow, 2009*). Accumulation of Notch receptor in endosomes, caused by mutations in genes involved in endo-lysosomal trafficking, leads to ectopic activation of Notch signaling (*Fortini and Bilder, 2009*; *Furthauer and Gonzalez-Gaitan, 2009*). Several proteins that are involved in endo-lysosomal trafficking, such as endosomal sorting complex required for transport (ESCRT) complexes, lethal (2) giant discs (lgd), the homotypic fusion and vacuole protein sorting (HOPS) complex, and the AP-3 complex, have been shown to regulate Notch signaling in flies (*Jaekel and Klein, 2006*; *Moberg et al., 2005*; *Thompson et al., 2005*; *Vaccari and Bilder, 2005*; *Vaccari et al., 2008*, *2009*; *Wilkin et al., 2008*). To our knowledge, this is the first report in which developmental defects of these endo-lysosomal trafficking complexes in vertebrates have been attributed to upregulated Notch signaling. Interestingly, loss of lgd function leads to ectopic upregulation of Notch signaling, and lgd specifically acts on Notch endo-lysosomal trafficking (*Jaekel and Klein, 2006*). We could speculate that BLOS2 may play a similar role in association with lgd by functioning as an adaptor of ESCRTs in vertebrates. BLOS2 is a shared subunit of BLOC-1 and BORC (*Pu et al., 2015*). Although BLOC-1 and BORC have been implicated in endo-lysosomal trafficking (*Cai et al., 2010*; *Ji et al., 2009*; *Pu et al., 2015*; *Zhang et al., 2014a*), our results have shown that BLOS2 probably regulates Notch signaling independently of BLOC-1 or BORC. Loss of BLOS2 leads to the accumulation of Notch in the endosomes. In comparison, when another BLOC-1 subunit (BLOS1 or snapin) is deficient, EGFR or D2R is accumulated in the plasma membrane, where EGF or D2 signaling is upregulated in a ligand-depend manner (*Supplementary file 1*) (*Cai et al., 2010*; *Ji et al., 2009*; *Zhang et al., 2014a*). The interesting point here is that the mechanism utilized by BLOS2 specifically regulates Notch1, but is not a general mechanism for lysosomal degradation of other endocytosed receptors, such as EGFR. There could be several hypotheses to explain this. First, different interacting partners of BLOC-1 subunits operate to expand their involvement in different pathways. Our recent results have shown that different subunits of BLOC-1, such as dysbindin, muted, and BLOS1, may have unique functions in lysosomal trafficking (e.g. in endo-lysosomal trafficking, LRO biogenesis, and autophagy) mediated by specific interacting partners (*Hao et al., 2015*; *Wang et al., 2014*; *Yuan et al., 2015*; *Zhang et al., 2014a*). In addition, BLOS2 may have a function in the nucleus (*Sun et al., 2008*). Second, the residual activity of the destabilized subunit outside of BLOC-1 may function in some way that is different from a null mutation itself, suggesting that phenotypes are differentially sensitive to genetic dosages of loss-of-function BLOC-1 alleles (*Mullin et al., 2015*).

HSPCs are capable of self-renewal and differentiation into all lineages of blood, including erythroid, myeloid and lymphoid lineages. HSPCs are now widely used in transplantation to cure multiple diseases. Inability of the donors to provide sufficient HSPCs makes it urgent to produce HSPCs ex vivo or in vitro. Our results revealed the inhibitory role of BLOS2 in HSPC production. Manipulations of BLOS2 expression in HSPCs may therefore have practical implications. Downregulation of

BLOS2 by genome-editing of HSPCs might facilitate its production, which is highly demanded in clinical applications. By contrast, overexpression of BLOS2 promoted HSPC differentiation, which may improve the differentiation efficiency. Interestingly, the differentiation activity of excessive HSPCs was abnormal as they generated more T cells at the expense of erythroid and myeloid differentiation. This phenotype is similar to T-cell acute lymphoblastic leukemia (T-ALL), which is also caused by the hyperactivation of the Notch signaling pathway (*Tzoneva and Ferrando, 2012*). The mimicking T-ALL phenotypes in BLOS2 knockouts have provided potential zebrafish or mouse models of T-ALL for clinical investigation. In addition, modulation of BLOS2 or the lysosomal degradation pathway may serve as a potential therapeutic target in controlling T-ALL progression in patients.

# Materials and methods

## Animals and genotyping

*Bloc1s2* knockout mice (*Bloc1s2$^{-/-}$*) were generated by gene targeting in 129/J-derived embryonic stem cells and bred in C57BL/6J at the animal facility of the Institute of Genetics and Developmental Biology (IGDB), Chinese Academy of Sciences. Other mouse mutants (*Pldn$^{pa}$*, *Dtnbp1$^{sdy}$*, *Kxd1*-KO, *Bloc1s1*-KO) used in this study have been described previously (*Huang et al., 1999*; *Li et al., 2003b*; *Yang et al., 2012*; *Zhang et al., 2014a*). All animal procedures were approved by the Institutional Animal Care and Use Committee of IGDB (KYD2006-002). WT and mutant *Bloc1s2$^{-/-}$* alleles were identified using the following primers: 5'-TGG GCT TAC AGA TGT TGT ATC AC-3', 5'-GCC AAG ATG CAG ACT CAG TAG CAG-3' and 5'-TGG GCT TAC AGA TGT TGT ATC AC-3'.

Zebrafish strains including Tubingen, *cmyb*:GFP (*North et al., 2007*), *kdrl*:mCherry (*Bertrand et al., 2010*), *fli1a*:EGFP (*Lawson and Weinstein, 2002*), *tp1*:dsRED (*Parsons et al., 2009*) and *bloc1s2$^{-/-}$* mutants were raised and kept at 28.5°C in the standard circulating water system. The embryos were obtained by natural spawning. This study was approved by the Ethical Review Committee in the Institute of Zoology, Chinese Academy of Sciences, China.

## Immunohistochemistry and immunofluorescence staining

Embryonic brains were dissected and fixed by immersion with 4% PFA and cryoprotected with 30% sucrose at 4°C. Tissues were embedded in Tissue-Tek OCT (Sakura Finetek) and sectioned on a Cryostat (Leica CM1900). For immunohistochemistry, coronal sections of 40 µm thickness were incubated with blocking solution containing 0.01 M phosphate-buffered saline (PBS) supplemented with 5% goat serum, 0.3% Triton X-100 and 3% bovine serum albumin (BSA). Primary antibodies were diluted in blocking solution and incubated overnight at 4°C. After washing, sections were incubated with fluorochrome-conjugated secondary antibody for 2 hr at room temperature. Finally, the sections were incubated with DAPI and mounted. For BrdU/Ki67 immunostaining, denaturation of DNA is required. Sections were incubated in 2M HCl for 30 min at 37°C, followed by neutralization in 0.1 M sodium tetraborate buffer (pH 9.0) for 10 min at room temperature. For Nissl staining, serial frozen sections were stained with 0.1% cresyl violet and then dehydrated with graded alcohols.

For immunofluorescence staining, cells were fixed in 4% paraformaldehyde (PFA) for 30 min and washed three times in PBS. Cells were blocked for 1.5 hr in blocking solution (PBS containing 3% BSA and 0.1% Triton X-100). Cells were then incubated at 4°C overnight with primary antibodies diluted in blocking solution, washed and incubated with secondary antibodies for 2 hr at room temperature. After washing, cells were incubated with DAPI and mounted. Primary antibodies used were as follows: rat anti-Ctip2 (1:500, Abcam, ab18465), rabbit anti-Tbr1 (1:200, Abcam, ab31940), rat anti-BrdU (1:100, AbCam, ab6326), rabbit anti-Ki67 (1:500, AbCam, ab15580), mouse anti-Tuj1 (1:1000, Abcam, ab7751), rabbit anti-Pax6 (1:500, Covance Research Products, PRB-278P), rat anti-PH3 (1:1000, Abcam, ab10543), rabbit anti-Notch1 (1:400, Cell Signaling Technology, 3608), mouse anti-EEA1 (1:100, BD Biosciences, 610457), mouse anti-LBPA (1:200, Echelon, Z-PLBPA), mouse anti-LAMP1 (1:500, BD Biosciences, 553792).

The mouse embryos were processed for cryosectioning (10 µM) and the immunofluorescence assay performed as described previously. Briefly, *Bloc1s2$^{+/+}$* and *Bloc1s2$^{-/-}$* embryos at E11 were fixed with 4% paraformaldehyde in phosphate-buffered saline (PBS) for 12 hr at 4°C. The sections were blocked in 5% BSA containing 0.3% Triton X-100 for 1 hr, and then incubated with Runx1 (Abcam, ab92336) and CD31 (Abcam) antibody overnight at 4°C and then with Alexa 488- and 561-

labeled secondary antibody (Invitrogen) for 2 hr. After that, the sections were counterstained with DAPI and all fluorescence images were acquired by Nikon confocal A1 (Nikon, Japan). BrdU and PH3 assays in zebrafish were performed as described previously (*Lu et al., 2015*; *Wang et al., 2013*).

## Immunoblotting

Tissues of $Bloc1s2^{-/-}$ mice and their WT littermates were dissected and homogenized in RIPA buffer (150 mM NaCl, 1% Nonidet P-40, 0.5% DOC, 0.1% SDS, 50 mM Tris, pH 8.0) and the protein content was determined with Protein Assay (Bio-Rad). The dissected trunk regions of zebrafish embryos were homogenized with cell lysis buffer containing proteinase inhibitors. Lysates were separated in SDS-PAGE gels and immunoblotted with the indicated antibodies. Rabbit BLOS2 polyclonal antibodies were generated against a His-tagged fusion protein corresponding to the mouse full-length $Bloc1s2$ gene. Other antibodies used in this study were as follows: rabbit anti-Notch1 (1:2000, Cell Signaling Technology, 3608), rabbit anti-Fabp7 (1:5000, Millipore, ABN14), rat anti-Hes1 (1:1000, MBL, D134-3), rabbit anti-Hes5 (1:1000, Millipore, AB5708), mouse anti-EEA1 (1:1000, BD Biosciences, 610457), mouse anti-CD63 (1:1000, Millipore, NG1944051), mouse anti-GM130 (1:1000, BD Biosciences, 610822), mouse anti-Flag (1:5000, Sigma, F3165), rabbit anti-Myc (1:5000, Sigma, C3956), mouse anti-β-actin (1:10000, Sigma, A5441), anti-Runx1 (1:200, AnaSpec, 55593), anti-Notch1 (1:200, Abcam, ab65297), anti-EGFR (1:5000, Fitzgerald, 20R-ER004), anti-Cathepsin D (1:1000, SantaCruz, sc-6486). Blots from three independent experiments were quantified using ImageJ (NIH) and represented by relative intensity versus that of control. β-actin was used as an internal control to normalize band intensity.

## Quantitative real-time polymerase chain reaction (qRT-PCR) analysis

Total RNA was extracted from WT and $Bloc1s2^{-/-}$ dissected cortices at E14.5 with RNeasy (Qiagen) and used to generate first-strand cDNA with the Superscript III cDNA synthesis system (Invitrogen) according to the manufacturer's instructions. qRT-PCR analysis was performed using SYBR PrimeScript Ready Mix (Takara) in an ABI 7900 sequence detection system (Applied Biosystems). GAPDH expression was used for normalization. Total RNAs are from the dissected trunk region of zebrafish embryos. The PCR primers are listed in *Supplementary file 2*.

## Cell culture, transfection and co-immunoprecipitation assay

WT and $Bloc1s2^{-/-}$ MEFs were isolated from each of the E14.5 embryos. MEFs, HEK293T or HeLa cells were all cultured in DMEM (Thermo Scientific) supplemented with 10% FBS (Hyclone Laboratories, Logan, UT) and 1% penicillin/streptomycin at 37°C, 5% $CO_2$. Ultrastructures of MEFs were examined by transmission electronic microscopy by following procedures described before (*Zhang et al., 2014a*).

WT and $Bloc1s2^{-/-}$ neurospheres derived from E14.5 NPCs were cultured as described previously (*Louis and Reynolds, 2005*). Neurospheres were cultured in DMEM/F12 medium with 1% B27 supplement (Gibco), bFGF (20 ng/ml, R and D), EGF (20 ng/ml, R and D) and 1% penicillin/streptomycin at 37°C, 5% $CO_2$. For immunofluorescence staining, mouse NPCs were cultured in dishes pre-coated with poly-D-lysine and in the DMEM/F12 medium with 1% B27 supplement (Gibco), bFGF, EGF and 1% penicillin/streptomycin.

Plasmids were transfected into HEK293T and HeLa cells with Lipo2000 (Invitrogen) and into MEFs with MACSfectin Reagent (Miltenyi Biotec) according to the manufacturer's protocol. The coding sequence of BLOS2 was amplified by PCR from mouse brain cDNA, and cloned into the pCMV-tag2B vector (with Flag-tag) and pCMV-tag3B vector (with Myc-tag). The pCS2 Notch1 Full Length-6MT (plasmid 41728) and 3 × Flag-NICD1 (plasmid 20183) were purchased from Addgene. Co-immunoprecipitation (Co-IP) assays were performed as described by *Zhang et al. (2014b)*. Transfected HEK-293T cells were harvested and lysed in 50 mM Tris-HCl (pH 7.4), 150 mM NaCl, 1 mM EDTA, 1% Triton X-100 and protease inhibitors. The lysate was incubated overnight with monoclonal anti-FLAG M2 antibody (Sigma) at 4°C. After washing, the samples were eluted with elution buffer (5 µg/µL 3 × FLAG peptide) and immunoblotted with anti-Myc or anti-Flag antibody.

The procedure of endogenous co-IP was performed as reported previously (*Zhang et al., 2015*). Briefly, E11 mouse AGM regions were dissected and lysed in RIPA buffer and incubated with Notch1

(Abcam, ab128076, 1:100) antibody for 4 hr at 4°C. Protein A+G agarose (Santa, SC-2003) was added next to the mixture which was incubated for another 3–4 hr. Beads were collected for western blotting using Notch1 and rabbit BLOS2 polyclonal antibodies.

## BrdU labeling, cell cycle analysis and TUNEL assay

Embryonic brains were collected 2 hr after BrdU (Sigma) injection (50 mg/kg body weight) into pregnant dams at E13.5. BrdU/Ki67 immunostaining was performed as described above. We calculated proliferation index as $(BrdU^+ Ki67^+)/Ki67^+$ where Ki67 marks all cycling cells. To assess cell cycle exit, we injected BrdU intraperitoneally into pregnant dams 18 hr prior to embryo harvest at E13.5–E14.5, and the cell-cycle exit index was calculated as $(BrdU^+ Ki67^-)/BrdU^+$ (*Siegenthaler et al., 2009*).

To analyze their cell cycle status, HSPCs were stained with antibodies against c-Kit-APC and CD34-PE (eBioscience) followed by fixation and permeabilization with Foxp3/Transcription Factor Staining Buffer Set (eBiosciences) according to the manufacturer's protocol. After that, the cells were incubated with Ki67-FITC (eBioscience) and stained with 7-amino-actinomycin D (7AAD) (eBioscience). The cells were analyzed with the MoFlo XDP (Beckman Coulter).

The TUNEL assay was performed with the In Situ Cell Death Detection Kit, POD (Roche) according to the manufacturer's protocol.

## Flow cytometry

Single cells obtained from E11 AGMs were dissociated by collagenase. Then antibody staining was performed for 30 mins at 4°C using antibodies specific to c-Kit-APC, CD31-PE, CD41-FITC, CD45-APC, CD34-PE and TER119-PE-CY7 (eBioscience). The cells were analyzed with the MoFlo XDP (Beckman Coulter).

## Colony-forming unit-cell assay

Colony-forming unit-cell (CFU-C) assay was performed as following. Each well that was plated with 600 $c-Kit^+CD34^+$ HSPCs from E11 $Bloc1s2^{+/+}$ and $Bloc1s2^{-/-}$ embryos was carefully harvested to ultra-low attachment 24-well plates (Costar) and cultured in CFU-C media. The cells were incubated at 37°C in 5% $CO_2$ with 100% humidity for 7–10 days, and the number of each type of colony was counted according to morphology. The experiment was repeated in triplicate.

## mRNA microinjection and whole-mount in situ hybridization

For mRNA synthesis, zebrafish *bloc1s2* full-length CDS were cloned into the pCS2+ vector. Capped mRNA was synthesized using the mMessage mMachine SP6 kit (Ambion) and purified using an RNA purification kit (Tiangen, China). *bloc1s2* mRNA were injected into 1 cell-stage wildtype and $bloc1s2^{-/-}$ embryos at the yolk/blastomere boundary.

Whole-mount in situ hybridization for embryos was performed using a ZF-A4 in situ hybridization machine (Zfand, China) as described previously (*Wang et al., 2011*), with probes including *bloc1s2*, *runx1*, *cmyb*, *rag1*, *gata1*, *pu1*, *hey2* and *ephrinB2a*.

## Pharmacological treatments

To block lysosomal or proteasomal degradation, MEFs were treated with the proteasomal inhibitors MG132 (10 μM) or the lysosomal inhibitors leupeptin (100 μM) for 4 hr. Cells were lysed in 50 mM Tris-HCl (pH 7.4), 150 mM NaCl, 1 mM EDTA, 1% Triton X-100 and protease inhibitors. The cell lysates were then subjected to SDS-PAGE and immunoblotted with the indicative antibodies.

To block Notch signaling, WT and $Bloc1s2^{-/-}$ MEFs were incubated with a γ-secretase inhibitor DAPT (25 μM, Sigma, D5942) for 5 hr, and *Hes1* expression was detected by qRT-PCR analysis. WT and $Bloc1s2^{-/-}$ neurospheres were treated with DAPT (1 μM) for 72 hr (*Hoeck et al., 2010*).

WT and $bloc1s2^{-/-}$ zebrafish embryos at 18 hpf were treated with a γ-secretase inhibitor DBZ (4 μM, Sigma, SML0649), MG132 (10 μM, Sigma, C2211) or chloroquine (50 μM, Sigma, C6628) respectively.

One embryo equivalent (ee) cells in AGM regions were treated with DMSO or DBZ (4 μM; Sigma, SML0649). The cells were incubated at 37°C in 5% $CO_2$ with 100% humidity for 4 d. Semi-adherent

cells were then carefully harvested and the antibody staining was performed as described previously, using antibodies specific to c-Kit-APC and CD45-PE (eBioscience).

## High-resolution imaging

Photomicrographs of Nissl-stained sections were obtained on a Leica SCN400 Brightfield scanner and confocal images were acquired with an ECLIPSE Ti-C2 confocal microscope equipped with NIS-Elements argon software (Nikon, Japan). We also performed structured illumination microscopy (SIM) on an OMX V4 system (DeltaVision OMX microscope, GE Healthcare). Images were further analyzed with Image-Pro Plus and ImageJ software. For morphological analysis, equivalent sections of $Bloc1s2^{-/-}$ and WT littermate brains were chosen on the basis of common morphological landmarks. Total thickness of the cerebral cortex, and thickness of the CP, IZ, and VZ were measured in Nissl-stained coronal sections of forebrains with Image-Pro Plus software. To calculate LV and brain volumes, the area from each serial section was estimated and volumes were estimated by summing the areas across sections and multiplying by slice thickness. For in vivo cell counting, numbers of marker-positive cells were counted and normalized to 100 µm of apical surface, defining a radial unit in more than four serial sections from at least three different animals per group. For Notch1 immunostaining, mean fluorescence values were measured in the VZ area and fluorescence background was subtracted in at least four serial sections from more than three different animals per group. For colocalization analysis in MEFs in vitro, at least 100 cells were counted in at least five different fields from at least three independent experiments. Mander's overlap coefficient K1 was calculated with NIS-Elements argon software. The zebrafish samples were prepared as previously reported (*Renaud et al., 2011*) and images were captured by Nikon A1 confocal microscopy (Nikon, Japan).

## OptiPrep gradient assay

The fractionation assay was performed using the OptiPrep gradient method as described by *Zhang et al. (2014b)*. The dissected WT and $Bloc1s2^{-/-}$ brains at E14.5 were immediately homogenized with HB lysis buffer (250 mM sucrose, 20 mM Tris-HCl, pH 7.4, 1 mM EDTA). An equal volume of 20% and 5% Optiprep (Axis-Shield, Norway) were placed into the tube in sequence and the tissue lysate was placed onto the top of the gradient. The sample was centrifuged at 28,000 rpm (TLS-55, Beckman, USA) for 14 hr at 4°C. Eleven fractions were collected from the top and equal aliquots from each fraction were analyzed for immunoblotting.

## Statistical analysis

Statistical analyses were performed by two-tailed Student's t-test or analysis of variance, using SPSS 11.5. qRT-PCR results were normalized as described by *Park et al. (2015)*. Sample sizes are indicated in the figure legends. All data are presented as mean ± s.e.m. We chose p=0.05 as a cutoff for statistical significance. In the figures, we use the star signs to represent different ranges of p values, *p<0.05, **p<0.01, ***p<0.001.

## Acknowledgements

We are very thankful to Dr Richard T Swank who provided the $Pldn^{pa}$ and $Dtnbp1^{sdy}$ mutants for this study and proofread this manuscript. We thank Dr Xiang Gao's lab for generating the $Bloc1s2^{+/-}$ mice. We are thankful to Mr Feng Zhang for his technical support. We thank Dr Hua Han and Dr Zhiheng Xu for discussions and suggestions.

## Additional information

### Funding

| Funder | Grant reference number | Author |
|---|---|---|
| National Natural Science Foundation of China | 91539204 | Wei Li |
| National Natural Science Foundation of China | 31471333 | Xin He |

| National Natural Science Foundation of China | 31271570 | Feng Liu |
|---|---|---|
| National Natural Science Foundation of China | 31230046 | Wei Li |
| National Natural Science Foundation of China | 91332116 | Wei Li |
| Chinese Academy of Sciences | KJZD-EW-L08 | Wei Li |
| Ministry of Science and Technology of the People's Republic of China | 2010CB945300 | Feng Liu |
| Chinese Academy of Sciences | XDA01010110 | Feng Liu |
| National Natural Science Foundation of China | 31425016 | Feng Liu |
| Ministry of Science and Technology of the People's Republic of China | 2011CB943900 | Feng Liu |

The funders had no role in study design, data collection and interpretation, or the decision to submit the work for publication.

## Author contributions

WZ, Conception and design, Acquisition of data, Analysis and interpretation of data, Drafting or revising the article; QH, CZ, Acquisition of data, Analysis and interpretation of data, Drafting or revising the article; XH, ZC, Analysis and interpretation of data, Contributed unpublished essential data or reagents; FL, WL, Conception and design, Analysis and interpretation of data, Drafting or revising the article

## Author ORCIDs

Wei Li, http://orcid.org/0000-0001-7430-6019

## Ethics

Animal experimentation: All animal procedures were approved by the Institutional Animal Care and Use Committees of IGDB or IOZ, CAS (protocol number: KYD2006-002).

## Additional files

### Supplementary files

• Supplementary file 1. A comparison of phenotypes for BLOS2-KO, BLOS1-KO and Snapin-KO mouse embryos or MEFs.

• Supplementary file 2. The PCR primers used in quantitative PCR assay.

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
