## [Decision Letter]

Thank you for submitting your article "BLOS2 negatively regulates Notch signaling during neural and hematopoietic stem and progenitor cell development" for consideration by *eLife*. Your article has been favorably evaluated by Janet Rossant (Senior Editor) and three reviewers, one of whom, Amy J Wagers (Reviewer #1), is a member of our Board of Reviewing Editors.

The reviewers have discussed the reviews with one another and the Reviewing Editor has drafted this decision to help you prepare a revised submission.

Summary:

This manuscript examines the role of the endo-lysosomal factor BLOS2 in neural and hematopoietic progenitors during mouse and zebrafish development. The authors report perturbations in developmental neurogenesis and hematopoiesis in Blos1s2 KO mice that mimic Notch mutations, leading them to investigate the impact of BLOS2 on Notch signaling. In a heterologous cell system (MEFs), they demonstrate that proteosomal degradation of Notch is impeded in the absence of BLOS2, leading to enhanced Notch signaling. They also show that Notch1 and Blos2 are found in complex and can be colocalized in endosomal compartments. Based on these data and insights from previous publications in *Drosophila*, the authors propose that in the absence of BLOS2, Notch1 gets activated in a "ligand independent manner" in this compartment. Interestingly, the authors further examined mutant/KO mouse lines that are defective in other components of the BLOC-1 and/or BORC complex and find that these animals do not phenocopy the neuronal defect (as well as the Notch1 accumulation phenotype in MEF cells) seen in BLOS2 KO mice. Based on these findings, the authors propose that the function of BLOS2 in Notch signaling is independent of its proposed function in the BLOS1-1/BORC complex.

The manuscript includes a substantial amount of data and reveals a new regulator of neural and hematopoietic progenitor cells, as well as a new player in Notch signaling. It further confirms earlier evidence in invertebrate models that the endo-lysosomal system is a major regulator of Notch signaling outputs during development. As such, it is likely to be of broad interest and appropriate for publication in *eLife*; however, there are a few issues that must first be addressed:

Essential revisions, requiring experimental studies:

1) Whether blos2 activity is specific to Notch or part of a broader regulation of endo-lysosomal biogenesis is incompletely addressed. On the one end, Notch1 can be found in complex with Blos2 and on the other Notch up regulation is not seen in Blos1 mutants. This should be clarified experimentally. Possible experiments to address the question of Blos2 specificity include: 1-check localization and signaling output for a second signaling receptor known to rely on the endo-lysosome (e.g. EGFR). 2-look by ultrastructure (conventional plastic embedding EM) at whether MVBs are present in Blos2 mutant embryos or more simply MEFs. 3-use a lysotracker, DQ-red or Cathepsin D maturation assay to establish whether Bloc1s2 mutant MEFs have impaired lysosomal function per se. Blos2 overexpression could also be informative, if it produces lysosomal phenotypes that are opposite to the ones of ko cells. Controls here could be lysosomal inhibitors. Also the possibility that activity is due to the whole BLOC-1 complex but some redundancy in some of the components (but not in Blos2) should be considered.

2) Figure 5 analyzes heterozygous fish, rather than homozygous disrupted animals. Why use heterozygous animals? Were there issues with the homozygous fish, such as earlier lethality? An explanation of this aspect of the experimental design should be included, and data on levels of BLOS2 in the heterozygous fish, in comparison to KO fish, should be added to allow for proper interpretation of the results.

3) Use of MEFs to interrogate the mechanism of Notch regulation in the cortex/HSPCs carries the caveat that these are a heterologous cell type. This caveat should be explicitly mentioned in the Discussion. Also, the MEF data support their model, immunostaining experiments should be performed to confirm this mechanism in neural and hematopoietic cell types. In particular, the authors show that the total Notch1 level (which the authors refer to as Notch1(TM)) is increased in BLOS2 KO mouse tissues; however, they do not show the subcellular compartment in which Notch1 is elevated (Figure 3, Figure 4). Based on the MEF data (Figure 7), one would predict that this is in late endocytic compartment but there may be differences depending on the cell type, considering that Notch regulation is highly context dependent. The authors should perform similar staining in the brain and AGM to clearly demonstrate that Notch accumulates in the endolysosomal system in vivo.

---

## [Author Response]

*[…] Essential revisions, requiring experimental studies:*

*1) Whether blos2 activity is specific to Notch or part of a broader regulation of endo-lysosomal biogenesis is incompletely addressed. On the one end, Notch1 can be found in complex with Blos2 and on the other Notch up regulation is not seen in Blos1 mutants. This should be clarified experimentally. Possible experiments to address the question of Blos2 specificity include: 1-check localization and signaling output for a second signaling receptor known to rely on the endo-lysosome (e.g. EGFR). 2-look by ultrastructure (conventional plastic embedding EM) at whether MVBs are present in Blos2 mutant embryos or more simply MEFs. 3-use a lysotracker, DQ-red or Cathepsin D maturation assay to establish whether Bloc1s2 mutant MEFs have impaired lysosomal function per se. Blos2 overexpression could also be informative, if it produces lysosomal phenotypes that are opposite to the ones of ko cells. Controls here could be lysosomal inhibitors. Also the possibility that activity is due to the whole BLOC-1 complex but some redundancy in some of the components (but not in Blos2) should be considered.*

We have performed additional experiments to address these points.

A) Our results showed that no changes in EGFR levels in *Bloc1s2^-/-^* MEFs or in BLOS2 overexpressed MEFs (new Figure 8—figure supplement 1). In addition, no significant difference of EGFR levels in *Bloc1s2^-/-^* cortex was observed (new Figure 8—figure supplement 1). These results suggest that the mechanism utilized by BLOS2 specifically regulates Notch1, but is not a general mechanism for lysosomal degradation of other endocytosed receptors, such as EGFR.

B) We have checked the ultrastructures of the organelles in endo-lysosomal system in *Bloc1s2^-/-^* MEFs. We adopted the definition of these organelles as described before (Zhang et al., J Biol Chem, 2014). There was no significant difference in the morphology and number of MVBs or late endosomes. However, a significant increase of endolysosomes was observed, with unknown effects on lysosomal trafficking of Notch1. No apparent change in lysosomal number was observed (new Figure 8—figure supplement 3).

C) We did not see apparent changes in mature and immature cathepsin D levels in *Bloc1s2^-/-^* MEFs, suggesting that the lysosomal function in *Bloc1s2^-/-^*MEFs was unaltered (new Figure 8—figure supplement 1).

D) In BLOS2-overexpressed MEFs, our results showed reduced Notch1 level (new Figure 3—figure supplement 2), which is opposite to the observation in *Bloc1s2^-/-^* MEFs.

E) In our control experiments, we did observe an increased EGFR level in leupeptin-treated MEFs (new Figure 8—figure supplement 1), suggesting our testing system was working.

In summary, our results suggest that BLOS2 functions specifically in the lysosomal degradation of Notch1. This function is independent of BLOC-1 or BORC as we have shown in Figure 8.

*2) Figure 5 analyzes heterozygous fish, rather than homozygous disrupted animals. Why use heterozygous animals? Were there issues with the homozygous fish, such as earlier lethality? An explanation of this aspect of the experimental design should be included, and data on levels of BLOS2 in the heterozygous fish, in comparison to KO fish, should be added to allow for proper interpretation of the results.*

We apologize for the unclear description of these results. The *bloc1s2* homozygous fish was viable which was used in this study. The ‘heterozygous fish’ was a mistake in the description of a transgenic line. We outcrossed *bloc1s2* homozygous fish with Tg (*flk1:mcherry;cmyb:*GFP) or Tg (*fli1a:EGFP;tp1:*dsRed) transgenic lines and finally got *bloc1s2* homozygous fish in two transgenic backgrounds. We repeated these imaging results in these *bloc1s2* homozygous embryos, as shown in Figure 5 and Figure 5—figure supplement 3, the numbers of cmyb^+^ and tp1^+^ cells in the dorsal aorta were increased in *bloc1s2* homozygous embryos. We also compared the protein levels of BLOS2 in the heterozygous and homozygous fish, and as shown in new Figure 5—figure supplement 2, BLOS2 was decreased but not diminished in the homozygous mutant fish. The explanation to the presence of BLOS2 protein is likely due to the encoded protein of *bloc1s2*-201 transcript (see details in the new Figure 5—figure supplement 2 to C). Therefore, the homozygous fish is a hypomorphic mutant. Please note that we have changed the name of homozygous fish (*bloc1s2^-/-^)* to *bloc1s2* hypo in the revised Figure 5, Figure 5—figure supplement 1 and Figure 5—figure supplement 3, as well as in the text.

*3) Use of MEFs to interrogate the mechanism of Notch regulation in the cortex/HSPCs carries the caveat that these are a heterologous cell type. This caveat should be explicitly mentioned in the Discussion. Also, the MEF data support their model, immunostaining experiments should be performed to confirm this mechanism in neural and hematopoietic cell types. In particular, the authors show that the total Notch1 level (which the authors refer to as Notch1(TM)) is increased in BLOS2 KO mouse tissues; however, they do not show the subcellular compartment in which Notch1 is elevated (Figure 3, Figure 4). Based on the MEF data (Figure 7), one would predict that this is in late endocytic compartment but there may be differences depending on the cell type, considering that Notch regulation is highly context dependent. The authors should perform similar staining in the brain and AGM to clearly demonstrate that Notch accumulates in the endolysosomal system in vivo.*

We have performed additional experiments to address this point:

A) In cultured neurospheres, we observed increased colocalization of Notch1 with LBPA (late endosome marker), but decreased colocalization of Notch1 with LAMP1 (lysosome marker) (new Figure 3—figure supplement 2).

B) Similar to the observation in neurospheres, we have found increased colocalization of Notch1 with LBPA and decreased colocalization of Notch1 with LAMP1 in the AGM region (new Figure 4—figure supplement 1).